



# The highest methane concentrations in an Arctic river are linked to local terrestrial inputs.

Karel Castro-Morales[1*], Anna Canning[2], Sophie Arzberger[1], Will A. Overholt[1], Kirsten Küsel[1,3], Olaf Kolle[4], Mathias Göckede[4], Nikita Zimov[5], and Arne Körtzinger[2,6]

[1] Friedrich-Schiller University Jena, Institute of Biodiversity, Jena, Germany.
[2] GEOMAR Helmholtz Centre for Ocean Research Kiel, Kiel, Germany.
[3] German Centre for Integrative Biodiversity Research (iDiv) Halle-Jena-Leipzig, Germany
[4] Max Planck Institute for Biogeochemistry, Jena, Germany
[5] Pleistocene Park, Northeast Science Station, Chersky, Russia.
[6] Christian Albrecht University Kiel, Kiel, Germany

**Correspondence:** Karel Castro-Morales (karel.castro.morales@uni-jena.de)

## Abstract.

Large amounts of methane ($CH_4$) could potentially be formed as a result of the gradual or abrupt thawing of Arctic permafrost due to global warming. Upon its release, this potent greenhouse gas can be emitted into the atmosphere, or transported laterally into aquatic ecosystems via hydrologic connectivity at surface or groundwaters. While high northern latitudes contribute up to 5 % of total global $CH_4$ emissions, the specific contribution of Arctic rivers and streams is largely unknown. In this study, we measured high-resolution continuous $CH_4$ concentrations in a ~120 km section of the Kolyma River in Northeast Siberia navigated twice between 15-17 June 2019 (late freshet). The average partial pressure of $CH_4$ ($pCH_4$) in tributaries (66.8 – 206.8 µatm) was 2-7 times higher than in the main river channel (28.3 µatm). In the main channel, $CH_4$ was up to 1600 % supersaturated with respect to atmospheric equilibrium. At key sites located near the riverbank and tributary confluences, $pCH_4$ (41±7 µatm) and emissions (0.03±0.004 mmol m$^{-2}$ d$^{-1}$) were higher compared to other sites within the main channel. Warm waters ($T$>14.5 °C) and low specific conductivities (κ<88 µS cm$^{-1}$) defined these key sites. The distribution of methane in the river could also be linked statistically to $T$ and κ of the water, as well as to the distance to the shore $z$, as indicators used to predict $CH_4$ concentrations in unsampled river areas. Similarly, the abundance of methane consuming bacteria and methane producing archaea strongly correlated mainly to $T$ and κ, and less to the $pCH_4$, and were similar to those previously detected in nearby soils, suggesting the source of $CH_4$ to be associated with sites close to land. The average total $CH_4$ flux densities in the investigated Kolyma River section were 0.02±0.006 mml m$^{-2}$ d$^{-1}$, equivalent to a total $CH_4$ flux of 12.4 mmol m$^{-2}$. Key sites with highest $CH_4$ concentrations contributed from 13 to 20 % to the total flux. Our study highlights the importance of high-resolution continuous $CH_4$ measurements in Arctic Rivers for identifying spatial and temporal variabilities, and offers a glimpse to the magnitude of riverine methane emissions in the Arctic and their potential relevance to regional methane budgets.



## 1 Introduction

Methane ($CH_4$) is a powerful greenhouse gas that absorbs the Earth's infrared radiation more efficiently than $CO_2$, with a global warming potential 28 times that of $CO_2$ over a time

horizon of 100 years (Saunois et al., 2020). To date, methane has accounted for 16 to 25 % of the current atmospheric warming (Etminan et al., 2016; IPCC, 2014; Rosentreter et al., 2021). Globally, aquatic ecosystems contribute about half (53 %) of the total $CH_4$ emissions, both from anthropogenic and natural origin (Rosentreter et al., 2021). The total bottom-up (i.e., from process-based models and inventories) updated global $CH_4$ emissions from rivers and

streams have a mean of 30.5±17.1 Tg $CH_4$ $yr^{-1}$ (Rosentreter et al., 2021), and account for ~17 % of the average inland water $CH_4$ fluxes (Saunois et al., 2020). Especially on regional scales, $CH_4$ emissions from rivers and streams have large impacts on the estimation of local atmospheric emissions (Karlsson et al., 2021). The contribution of $CH_4$ emissions in high northern latitudes (60 – 90° N) to total global $CH_4$ emissions ranges between 4 to 5 %, but

there are significant uncertainties, particularly regarding the contributions from terrestrial permafrost and non-wetland inland waters, i.e., rivers, streams, and lakes (Saunois et al., 2020). The concentration of $CH_4$ in rivers and streams is generally above saturation with respect to the present atmospheric $CH_4$ concentration, emitting annually the equivalent to ~15 % of the total emissions from wetlands or 40 % of the annual $CH_4$ emissions from lakes

(Stanley et al., 2016).

The Arctic Ocean is one of the most river-influenced and land-locked of all the world oceans (Charkin et al., 2017; Shakirov et al., 2020), receiving annually about 10 % of the global runoff (Lammers et al., 2001), through the input from the main six Arctic rivers: Yenisey, Lena, Ob, Mackenzie, Yukon, and Kolyma. These rivers connect the ocean with the land, by

mediating the transport of $CH_4$ stored in terrestrial surface waters or groundwaters, or through soil-water interactions in thawed water tracks (Connolly et al., 2020; Dabrowski et al., 2020; Harms et al., 2020; Saunois et al., 2020). Thus, the riverine transport of soil-derived $CH_4$ from permafrost may influence the $CH_4$ concentrations in the Arctic shelf system.

The atmospheric emissions of $CH_4$ from Arctic inland freshwaters and permafrost have the

potential to increase with climate change (Dean et al., 2018). As permafrost thaws, more soil organic carbon is available for the anaerobic degradation of organic matter under warmer conditions, resulting in additional $CH_4$ formation of which will add to the positive feedback to climate change (Schuur et al., 2015). Trapped or newly formed $CH_4$ can be emitted directly to the atmosphere after the abrupt or gradual permafrost thaw (Olefeldt et al., 2013; Saunois et

al., 2020; Turetsky et al., 2020), or be laterally transported into neighboring inland waters via



surface hydraulic connectivity or underground drainage (e.g., Dabrowski et al., 2020). Current and projected changes in the Arctic land surface hydrology, vegetation, landscape, and temperature due to permafrost thaw, will modulate $CH_4$ concentrations in Arctic fluvial ecosystems (Harms et al., 2020; Olid et al., 2021).

The magnitude of the fluvial $CH_4$ emissions is subject to strong local environmental controls, because $CH_4$ has low solubility in water (Campeau and del Giorgio, 2014; Stanley et al., 2016). At the same time, the abundance and phylogenetic identity of microorganisms in the river water that can be associated to the formation or consumption of $CH_4$, can serve as indicators of the source and fate of $CH_4$ transported from land. Aquatic $CH_4$ is subject to

microbial oxidation and photochemical decomposition (Dean et al., 2018; Stanley et al., 2016). Little is known about the magnitude of $CH_4$ concentrations and emissions from flowing Arctic inland waters, as well as how they vary over time and space. Point $CH_4$ measurements in some Arctic rivers and streams have demonstrated supersaturation relative to the atmosphere (e.g., Kling et al., 1992; Mann et al., 2022; Striegl et al., 2012; Vorobyev et

al., 2021; Zolkos et al., 2019). However, highly resolved aquatic $CH_4$ measurements are lacking in large portions of Arctic rivers and streams, and these are needed to better quantify the atmospheric gas fluxes and understand the temporal variations and the environmental indicators. High-resolution measurements of the partial pressure of methane ($pCH_4$) were measured in a site in Ambolikha River, a tributary of Kolyma River in northeast Siberia,

evidencing aquatic $CH_4$ supersaturations up of the order of 200 times higher than values at equilibrium with the atmosphere. These measurements allowed identifying temporal variations mostly driven by hydrological changes and air-water exchange, with a consistent decrease of $pCH_4$ by 78 % from the measured concentrations during late freshet to summer (Castro-Morales et al., 2022).

Here, we present the first high spatial resolution measurements of $pCH_4$, and other complementary water properties, in a large section of the Kolyma River during the late freshet (June) in 2019. Additionally, we followed the riverine microbial community structure using a 16S-amplicon approach along the same 120 km long transect, to provide a potential record of water input sources. The objectives of this study are: 1) to analyze potential environmental

indicators that can be statistically associated with the spatial variations of the $pCH_4$ along the sampled river section, 2) to estimate the flux of $CH_4$ across the atmosphere-river interface, and 3) to investigate a potential link between overall microbial community structure and more specifically the distributions of methane oxidizers and methane producer with the measured $pCH_4$ during the sampling period.



## 2 Methods

### 2.1 Study site and fieldwork description

The Kolyma River is the sixth largest river in the Arctic, with a watershed area of 653,000 km² (Holmes et al., 2012), that is completely underlain by continuous permafrost (Mann et al., 2012). Our area of study was a ~120 km section in the Kolyma River, bounded by the city of Chersky (68° 45' 5.1" N, 161° 18' 16.6" E) to the east and at the location known as Duvannyi Yar (68° 38' 12.8" N, 159° 5' 25.4" E) to the west (Fig. 1). Several floodplains are located next to the banks of this section of the Kolyma River. These floodplains connect the river to the land during the snow melt period (May and June) when they become inundated. We twice navigated the Kolyma River section onboard a small vessel (average navigation speed of $2.0 \pm 0.4$ m s$^{-1}$), where we installed our instruments for measurements of continuous water properties and the partial pressure of methane ($p$CH$_4$) (Sect. 2.2.). The first transect was navigated in the upstream direction (UP) from Chersky to Duvannyi Yar (Fig. 1) between 15 June 2019 (12:48 h; local Chersky time) and 16 June 2019 (16:59 h) (with an overnight break halfway), covering a length of 127.7 km. The second transect was navigated in the downstream direction (DOWN) from Duvannyi Yar to Chersky, and took place between 16 June 2019 (17:00 h) and 17 June 2019 (13:27 h), covering a length of 115.4 km.

In 2019, the ice break-up in Kolyma River at Chersky started on 1 June, and our sampling took place during the late freshet. Thus, during the sampling campaign the transect navigated was completely ice-free and in the decreasing phase of the freshet peak discharge as shown by the daily records from the gauge station Kolymsk-1 (68° 43' 48" N, 158° 43' 12" E) in the Kolyma River (Fig. S1). During the sampling days, the average width of the Kolyma channel was about 2 km. With help of the Arctic DEM Explorer (Environmental Systems Research Institute, Polar Geospatial Center; https://livingatlas2.arcgis.com/arcticdemexplorer/), we estimated a total area of the sampled Kolyma River section of about 221 km².

Continuous water properties were measured along both transects (Sect. 2.2). The vessel primarily navigated at the center of the Kolyma River main channel during the sampling, particularly in the DOWN transect. We purposely navigated in the proximity of the confluences of tributaries and in banks adjacent to floodplains during the UP transect to capture the water properties in regions with visually evident, large lateral contributions from land (i.e., runoff from land as evidenced by more turbid and/or differently colored water). To facilitate the analysis of the high-resolution data and analyze the specific contribution of banks and confluences with tributaries to the measured water properties and $p$CH$_4$, we defined five key sites (i.e., S1 to S5) that are associated with sampling points along the UP transect.



From east to west the location of the "key sites" is: S1, bank of floodplain 1 at Ambolikha
river in station PP07; S2, the confluence of tributaries Maly Anyuy and Bolshoy Anyuy
(M&B Anyuy) in station PP11; S3, bank of floodplain 2 (only in DOWN transect) in station
PP20; S4, bank of floodplain 3 (only in UP transect) in station PP23; and, S5, bank at
Duvannyi Yar in station PP25 (Fig. 1). The separation between "key sites" and the "other
sites" of the data was done on the basis of the measured $p$CH$_4$, $T$ and $\kappa$ in the UP and DOWN
transects, and also analyzed independently for each transect.

The UP and DOWN transects were not navigated exactly at the same locations and the
geographical overlap took place only in a few areas (Fig. 1). Therefore, we compare the
results between these transects in the context of the temporal variability of the measured
parameters, while the spatial variation is done between the key and other sites of the areas for
each transect.

## 2.2 Collection of discrete river water samples and analysis

During the UP transect, we collected discrete water samples at 21 sampling stations (PP05-
PP25) distributed along the track (see Fig. 1 for location and Table S1 for sampling times and
average water properties measured at each station), for the analysis of dissolved organic
carbon (DOC) and the composition of microbial communities (Sect. 2.3.). For this, a 1.5 L
Niskin bottle was lowered to 1 m depth and water samples were drawn from the sampler
onboard through silicone tubing.

### 2.2.1 Analysis of Dissolved Organic Carbon (DOC) in river water samples

For the quantification of DOC, a volume of 250 mL of water was transferred from the Niskin
bottle into an acid-washed amber glass flask for quantification of DOC. The samples were
stored at 4 °C until pre-treatment at the laboratory of the Northeast Science Station in Chersky
after the sampling campaign. The samples were brought to room temperature and filtered
through a pre-combusted 0.7 µm GF/F filter (Whatman®). Two aliquots of 10 mL of the
filtrate for each sample were transferred to acid-washed glass vials and acidified to pH 2.0
with 37 % HCl. The samples were kept cold during storage and transport to Germany for the
determination of DOC via high-temperature catalytic combustion (Analytik Jena), with each
sample measured from three to five times as analytical replicates.

### 2.2.2 Analysis of microbial communities in river water samples

We determined the distribution and total community composition of microbial communities,
including CH$_4$-producing archaea (methanogens) and CH$_4$-oxidizing bacteria (methanotrophs
and methylotrophs) in the river water samples. Methanotrophs utilize CH$_4$ as carbon source,
whereas methylotrophs are more versatile and can also use other C1 compounds as carbon



source. In addition, the abundance of bacteria and archaea was determined along the transects.
For this, a volume of 500 mL of the surface river water from the Niskin bottle was transferred
into a 500 mL glass flask (DURAN® Borosilicate glass, SCHOTT). Using a hand pump and
filtration system, this sample was immediately filtered on board through a 0.2 μm filter
(Supor®). The 500 mL were divided into three aliquots and filtered independently for
analytical replication. The filters were stored inside 2 mL sterile Biozym tubes and submerged
in DNA/RNA shield solution (Zymo Research). The tubes with the filters remained at room
temperature for their subsequent transport and analysis in Germany for DNA isolation,
amplicon sequencing, and 16S rRNA gene quantification following protocols specified in the
supplementing text S.1.2. and S.1.3.

### 2.3   Instrumental setup

Two instruments were installed onboard the vessel for continuous measurements of water
properties: (1) an EXO2 multiparameter sonde with seven sensors for simultaneous optical
and non-optical water measurements (Sect. 2.3.1), and (2) a Flow-Through (FT) system for
continuous measurements of the partial pressure of $CH_4$ ($pCH_4$) (Sect. 2.3.2). The instruments
were continuously fed with water pumped from the port side of the vessel from a nominal
depth of 1 m below the water surface, hereinafter referred to as "surface water". The surface
water was delivered through a PVC tubing of 2.5 m length and split into two outlets: 1) to
feed the FT system at an approximate flow rate of 0.14 L s$^{-1}$, and 2) to a 20-L FT box located
onboard where the EXO2 probe was immersed for the continuous surface water
measurements.

### 2.3.1   EXO2 Sonde

The EXO2 multiparameter sonde (YSI Inc., Xylem Inc., Yellow Springs, OH, USA) was used
to measure optically the turbidity (in formazin nephelometric units, FNU), dissolved $O_2$ (DO,
μmol L$^{-1}$), and fluorescent dissolved organic matter ($f$DOM; Quinine Sulfate Units, QSU) of
the incoming surface river water. It also measured temperature-corrected conductivity
(specific conductivity, $\kappa$ in μS cm$^{-1}$) with conductivity electrodes, water temperature ($T$, °C)
with a thermistor, and pH with a glass electrode. The sonde had an internal battery and was
mounted inside a metal frame (to provide protection and stability) submerged inside the 20 L
FT box that received the incoming water pumped from the surface. The bucket was kept
covered with a lid to avoid heating of the water and light exposure of the sensors. Considering
the same water flow rate at the FT box as in the FT system, the water retention time in the FT
box was on average 2.3 min, which allowed a sufficient time for the sensors in the probe to
stabilize for a reliable measurement.



The sonde was equipped with a wiper brush that was used routinely to clean the window of the sensors to avoid interferences due to fouling caused by the accumulation of deposits. The wiping periods were registered and removed from the data set. We obtained one measurement
every 5 sec and the data was monitored and stored in a computer onboard.

As a result of the travel distance of the pumped water through the pipe (see Sect. S.1.1 for details), the water within the 20 L bucket on board was on average 0.6 °C warmer and with 1.2 mg L$^{-1}$ higher DO content than the in-situ water at 1 m depth. Thus, the EXO2 sonde $T$ and DO measurements were corrected by these mean values. All the sensors of the sonde
were factory-calibrated previous to the measurements. Two-point calibrations were performed on-site to the DO and pH sensors and no analytical drift was observed before and after the measurements that would have required correction. The measured $f$DOM was temperature corrected to a reference of 25 °C (Downing et al., 2012; Watras et al., 2011), and further corrections due to the turbidity influence in the sensor response to light attenuation were done
after Snyder et al. (2018).

### 2.3.2   Flow-through (FT) system

The FT system is a portable and versatile flow-through sensor set-up for continuous direct measurements of $p$CH$_4$ from surface water. We used a HydroC® CH$_4$ FT sensor based on tunable diode laser absorption spectroscopy (-4H-JENA engineering GmbH, Jena,
Germany). A SBE45 thermosalinograph sensor (Sea-Bird Electronics, Bellevue, USA) was used to measure the temperature ($T\_{FT}$, °C) and conductivity of the incoming water. The HydroC® CH$_4$ FT sensor was factory-calibrated before and after the measurement campaign. The calibration and validation of the data were done following Canning et al. (2021a). Drift and response time corrections were not applied because we assume sufficient exposure of
the water to the sensor at the low sailing speeds. Because the relatively long response time of the CH$_4$ sensor (of the order of 20 min), the obtained data are significantly smoothed and therefore, the captured gradients and extreme values might not be precisely geographically located. However, the advantage of the high-spatial-resolution data allowed for a surface coverage that help identify high methane concentration areas. For more in-depth corrections
see Canning et al. (2021a).

Besides the slow navigation speed, the average time spent at each sampling station was 7±13 min (minimum of 2 min and maximum of 8 min), which allowed for further equilibration times of the surface water at the sensors of the instruments, particularly at sites with high methane concentration.

We obtained one measurement every 5 sec and the data were monitored and stored on a





computer onboard. The EXO2 sonde and FT system data were averaged to 1 min values.
During the measurements, we also navigated inside smaller tributaries, one located at
halfway along the transect length (named here as Leonid's stream), and another at the end
of the DOWN transect, located along the Ambolikha River. Because the water properties

measured in these streams are very contrasting to the properties in the main stem, we
removed these sections from the full data, but still present the average values measured along
those transects.

### 2.4    CH₄ flux calculation

To obtain the gas exchange across the water-air interface (i.e., flux density) it is necessary to

calculate the gas transfer velocities $k$. Here we followed two methods to obtain $k$: 1) using a
hydraulic model as a function of water velocity and discharge, and the river configuration
(Raymond et al., 2012), and 2) using a parameterization as a function of wind speed
(Wanninkhof, 2014). This was done in order to cover a range of values given the large
uncertainties of $k$ in rivers.

The hydraulic model that we used to calculate $k$, is a function of stream velocity ($V$, m s$^{-1}$),
river slope ($S$, unitless), water discharge ($Q$, m$^3$ s$^{-1}$), and water depth ($D$, m) (empirical Eq. 7
in Raymond et al., 2012):

$$k\_R12 = 4725 \times (VS)^{0.86} \times Q^{-0.14} \times D^{0.66} \qquad (1)$$

The average stream velocity for the transect ($V$=1.27±0.1 m s$^{-1}$) was calculated from the mean

daily water discharge from 15 to 17 June 2019 as reported at the gauge station Kolymsk-1
($Q$=13267±950 m$^3$ s$^{-1}$) divided by the mean cross-sectional area in the channel
($A$=10400±9721 m$^2$). $A$ was calculated from the average river depth ($D$=5.2±4.9 m) times the
river width ($W$ fixed at 2000 m) at the sampling times. The slope $S$ for the Kolyma River
along the 120 km channel was 0.003 % considering the mean elevation of 4 m, obtained from

the slope map in the Arctic DEM Explorer (Environmental Systems Research Institute, Polar
Geospatial Center; https://livingatlas2.arcgis.com/arcticdemexplorer/). An uncertainty of up to
7.8 % is obtained in this calculation mostly due to the use of an average river depth for the
calculation of the cross-sectional area and the stream velocity. The section of the Kolyma
River can be in places as shallow as 1.7 m and as deep as 21.6 m, leading to faster water

flows as the water column is shallow. However, larger uncertainties are expected due to the
variation in $Q$ along the stream, since the values used here are daily averages measured at
once single site at the Kolymsk-1 gauge station.

The empirical wind speed parameterization is used to also calculate $k$, followed by
Wanninkhof (2014):





$$k\_W14 = 0.251 \times (u_{10})^2 \qquad (2)$$

Were $u_{10}$ (m s$^{-1}$) is the wind speed normalized to 10 m above the water surface, following Amorocho and Devries (1980), calculated from the wind velocities measured at a height of 6 m above ground at a nearby eddy tower during the sampling period (Castro-Morales et al., 2022).

The $k\_R12$ from the hydraulic model and $k\_W14$ from the wind parameterization were standardized to a constant temperature using the Schmidt number ($Sc$) for CO$_2$ and freshwater at 20 °C, i.e., $Sc_{CO2}=600$ (Wanninkhof, 1992), and the $Sc$ of CH$_4$ ($Sc_{CH4}$) (Wanninkhof, 2014) following:

$$k_* = k\_* \times \left(\frac{Sc_{CH4}}{Sc_{CO2}}\right)^{-0.5} \qquad (3)$$

The water-to-air flux density of CH$_4$ ($F$, amount area$^{-1}$ time$^{-1}$) was obtained with the following function: $F_* = k_* \cdot \left(C_w - C_{eq}\right)$, where $k_*$ is the gas transfer velocity (length time$^{-1}$) of CH$_4$ at the in-situ $T$ (Eq. 3) for R12 or W14 (Eq. 1 and 2). The water-side equilibrium concentration of CH$_4$ ($C_{eq}$, μmol L$^{-1}$) is subtracted from the measured bulk CH$_4$ concentration in the water ($C_w$, μmol L$^{-1}$). $C_w$ was calculated from the Bunsen solubility coefficient ($\beta$, mol

L$^{-1}$ atm$^{-1}$) that is calculated as a function of temperature Weiss (1970), while $C_{eq}$ was calculated following Wiesenburg and Guinasso (1979). The atmospheric $p$CH$_4$ (atm) was calculated following:

$$p = x(P - pH_2O) \qquad (4)$$

where $x$ is the dry air mole fraction of CH$_4$. $P$ is the barometric pressure and $pH_2O$ is the

saturation water vapor pressure at in-situ water temperature (both in atm). We used the global mean dry air mole fraction of 1858.8 ppb for CH$_4$ during June 2019 according to the Global Monitoring Laboratory, NOAA (Dlugokencky and Tans, 2019), and a standard barometric pressure of 1 atm.

### 2.5 Data analysis

### 2.5.1 Correlation between water $p$CH$_4$ and water parameters

To simplify the analysis for finding the relationship between the multiple water parameters measured along the transect and $p$CH$_4$, we calculated 1-min averages from the continuous measurements of $T$, κ, pH, DO, $f$DOM and $p$CH$_4$ at the location of the discrete sampling stations in the UP transect. We also included for this analysis the DOC concentrations from

each station (average values are summarized in Table S1). In addition, we calculated the shortest distance from each station ($z_{stas}$) and of the navigated transects ($z$) to any of the river banks and considered this distance as another parameter relevant for the distribution of $p$CH$_4$



in the river. The river banks along the navigated transects were digitized in Google Earth
Pro®, and no other property was used to define the geographical location of these limits;

hence, the river banks are fixed locations without temporal variation for the period of our
sampling. We obtained $z_{stas}$ and $z$ from the shortest physical distance between the
geographical positions of the sampling stations and of the UP and DOWN transects to any of
the defined river banks. The river banks and limits of the transect define the polygonal area of
interest for this study (Fig. S2).

To find the correlation between the 1-min averages of $p$CH$_4$ and the water properties, as well
as the DOC at the sampling stations and $z_{stas}$, we performed a Pearson pairwise linear
correlation analysis ($p < 0.1$).

**2.5.2    Random forest regression analysis for extrapolation of transect $p$CH$_4$ into a
polygonal area of the river section as a function of $T$, $\kappa$ and $z$**

We estimated the $p$CH$_4$ at the sampling times in the entire river area of a polygon delimited
by the river banks and the limits of the navigated transects (Fig. S2). The river bank-forming
polygon of the Kolyma River section covered an area of 236.3 km$^2$. Within this area we
constructed a fine grid regularly distributed within the river polygon and with a horizontal
spatial resolution of 0.1 km.

We then built a fine grid polygon in the river for $T$ and $\kappa$ based on their best fit correlations to
$z$ at the transect scale, for the "key sites" and for the "other sites" (depending on the measured
$p$CH$_4$, $T$ and $\kappa$) at the sampling times during the UP and the DOWN transects. The gridded
products were used to extrapolate $p$CH$_4$ to other areas of the river as defined by the gridded
area delimited by the river banks, and on the basis of the highly spatially resolved $p$CH$_4$

measured along the transects.
For this, we obtained a best fit between $T$, $\kappa$ and $z$ to $p$CH$_4$ by applying a random forest
regression analysis. First, for "key sites", this was done as a function of $T$ and $\kappa$, i.e.,
$p$CH$_{4\_key}$($T_{key}$, $\kappa_{key}$). Second, for "other sites" it was done as a function of $T$, $\kappa$ and $z$, i.e.,
$p$CH$_{4\_other}$($T_{other}$, $\kappa_{other}$, $z$). These models were applied to the gridded polygon to extrapolate

$p$CH$_4$ from the transects to the entire gridded polygon. Once a gridded $T$ and $\kappa$ was obtained,
the corresponding model for $p$CH$_{4\_key}$ and $p$CH$_{4\_other}$ was applied. This procedure was done
independently for the UP and DOWN data.

**3    Results**

**3.1    Spatial distribution of continuous surface $p$CH$_4$ and water properties in UP and
345        DOWN transects**

The high-resolution continuous measurements of surface $p$CH$_4$ show significant spatial



heterogeneity and temporal variability in both the UP and DOWN transects (Fig. 2). Overall, high $p\mathrm{CH_4}$ (up to 46 µatm) was measured in the presence of warm (15.5 °C) and less conductive ($\kappa < 88$ µS cm$^{-1}$) water, and mostly located closer to the river banks ($z < 1.0$ km)

(Figs. 3 and 4).

During the UP transect, the average measured $p\mathrm{CH_4}$ was 25.8±6.7 µatm (or in terms of $\mathrm{CH_4}$ concentration, $C_\mathrm{w} = 41.5\pm9.2$ nmol L$^{-1}$). These values were measured in colder (by 0.6 °C) and less conductive waters (by 16.1 µS cm$^{-1}$) compared to the DOWN transect that was navigated two days later. The DOWN transect had on average 7.4 µatm higher $p\mathrm{CH_4}$

(33.2±9.4 µatm, or 54.3±14.7 nmol L$^{-1}$) than the UP transect (Table 1 and Fig. 3). In both transects, the concentration of $\mathrm{CH_4}$ remained supersaturated (by 1189±198 % in the UP transect and 1622±380 % in the DOWN transect) with respect to the concentration at atmospheric equilibrium (average 3.2±0.04 nmol L$^{-1}$).

The spatial distribution of water properties measured in both transects depicted evident

differences between the center of the main stem and the areas at the proximity of banks adjacent to floodplains and at confluences of tributaries with the Kolyma main stem (Fig. 2a and 2b, and supplementary Fig. S3). Specifically, hot spots of $p\mathrm{CH_4}$ with values > 35 µatm were measured in the key sites at the time of the measurements (Figs. 2 and 3).

During the UP transect, the maximum measured $p\mathrm{CH_4}$ was 46.1 µatm at site S5 (Duvannyi

Yar), very similar to the value measured during DOWN at the same location (i.e., 44.6 µatm). The maximum $p\mathrm{CH_4}$ measured in the main stem (80.7 µatm) was found at a site halfway along the DOWN transect in a site at the outlet of Leonid's stream (location 68.5281 °N, 160.3437 °E). However, the highest $p\mathrm{CH_4}$ was measured inside streams or tributaries with up to 222.9 µatm at Ambolikha River and up to 92.9 µatm inside Leonid's stream, both

navigated during the DOWN transect (Table 1). Larger supersaturations with respect to the atmospheric equilibrium were observed at these two transects with 9610±403 % in Ambolikha River and 3415±1051 % in Leonid's stream.

In addition to $p\mathrm{CH_4}$, $T$ and $\kappa$ were considered to distinguish between the key sites S1 to S5 from the other sites in the river. The key sites S1 to S5 were characterized (besides $p\mathrm{CH_4} > 37$

µatm) by the presence of warmer ($T > 14.5$ °C) and less conductive water ($\kappa < 88$ µS cm$^{-1}$) at the sampling time. Finally, because the key sites S1 to S5 were evidently located in the proximity of tributary confluences and banks (i.e., $z < 0.8$ km), we also considered $z$ (distance to the river bank) as a parameter related to high $p\mathrm{CH_4}$ in the main stem (Fig. S2 in supplement). The average, minimum and maximum values of $p\mathrm{CH_4}$, $C_\mathrm{w}$, $T$, and $\kappa$ in the UP

and DOWN transects at "key sites" and all "other sites" of the transect are summarized in



Table 1.

Other areas along the transects where $p$CH$_4$ was higher than 37 µatm were not included as part of the key sites because their corresponding $T$ or κ did not meet the properties specified above, e.g., at the site of the maximum $p$CH$_4$ of 80.7 µatm at the outlet of Leonid's stream

where the $T = 15.4$ °C and κ = 113.1 µS cm$^{-1}$ (Figs. 2 and 3).

The pairwise linear correlation analysis ($p < 0.1$) between all the measured parameters showed a statistically significant positive correlation between $p$CH$_4$ and $T$ ($r^2 = 0.51$), and a negative correlation to κ ($r^2 = 0.22$), $z_{stas}$ ($r^2 = 0.36$), and DO ($r^2 = 0.17$). No significant correlation was found between $p$CH$_4$, $f$DOM and turbidity.

To analyze if any of the measured water parameters had an influence on the distribution of $p$CH$_4$, we chose the conservative tracers to which CH$_4$ was significantly correlated: $T$, κ, and $z_{stas}$. These conservative parameters are then considered as potential predictors for the presence of dissolved CH$_4$ in the river, in contrast to reactive tracers such as DO that can be biologically or chemically altered in the river water. The analysis of environmental indicators

was done with the continuous high-resolution data only for the main stem areas.

### 3.2 Influence of conservative tracers on the distribution of riverine $p$CH$_4$ along transects and random forest regression as a gap-filling approach

The variations of $T$ and κ in the river are influenced by the proximity to the outlets of tributaries and the riverbanks. This influence is more evident in the UP transect, where $T$ and

κ at "key sites" correlated positively with $z$ ($r^2 > 0.45$, $p = 0.05$) (Fig. S4). In the data for the "other sites", the relation between $T$ and κ vs. $z$, followed a semi-logarithmic fit ($p = 0.05$) in both the UP and DOWN transects (Fig. S5).

To be able to fill gaps and extrapolate the $p$CH$_4$ measured along the transects into the entire polygonal river area, we employed a random forest regression approach based on the

correlations between $T$, κ and $z$. For this, we first built a fine-gridded polygon for $T$ and κ using the linear (for "key sites") and semi-logarithmic correlations (for "other sites") observed at the transect level during the sampling times. Once a gridded $T$ and κ were generated, the corresponding random forest model for $p$CH$_{4\_key}$ and $p$CH$_{4\_other}$ at transect level as a function of $T$, κ and $z$ correspondingly, was applied. This procedure was done independently for the

UP and DOWN transects, hence two polygons representing the modeled $p$CH$_4$ during 15–16 June 2019 and 16–17 June 2019 were obtained (Fig. S6).

To validate the output of the random forest models, we compared the measured and modeled $p$CH$_4$ along each transect. Results show that the skill of the model for the UP transect better reproduces the $p$CH$_4$ with an uncertainty of 3.9 µatm than that of the model for the DOWN



transect (uncertainty of 9.1 µatm) (Fig. S7). A larger error is observed in the areas of the key

sites mostly during the DOWN transect.

### 3.3    Microbial composition and DOC analysis in discrete water samples

Similar to the influences of temperature ($T$) and specific conductivity ($\kappa$) on the distribution

of $p$CH$_4$, we found that microbial community composition was significantly related to both $T$

(F = 15.5, $r^2$ = 0.17, $p$ < 0.001) and specific conductivity ($\kappa$) (F = 12.7, $r^2$ = 0.14, $p$ < 0.001)

(Fig. 5), while distance to the shore ($z$) was not significant. The $p$CH$_4$ measurements alone

explained a low portion of the community variance ($r^2$ = 0.06, p < 0.03), and when tested in

conjunction with both $T$ and $\kappa$, was not a significant contributor to microbial community

variance. In this way, microbial community composition can act as a record of $p$CH$_4$, as

microbes – and $T/\kappa$ – are less dynamic than $p$CH$_4$. Within the context of the strong patterns

related to both $T$ and $\kappa$, there were spatial patterns that reflected the location within the main

stem and the influences of tributaries, with key site S3 (PP20) exhibiting the lowest

similarities with the other four key sites and clustering with other water samples collected

within the main stem of the river. Conversely, key sites S1 and S2 clustered separately from

all other water samples, likely due to the heavy influence of tributary outflow and floodplain

inputs (Fig. 5).

Quantifying the 16S rRNA gene abundances of total archaeal and bacterial populations

revealed that archaea, were three orders of magnitude lower in abundance than their bacterial

counterparts across the river transect. However, the abundances of both were found to

strongly correlate (Pearson, $r^2$ = 0.81, $p$ < 1.8e–15) (Supplemental Fig. S8). Within the

archaeal 16S sequences detected, we found two putatively methanogenic OTU, each

belonging to a different family/genus (*Methanobacteriaceae* – *Methanobacterium*,

*Methanoregulaceae* – *Methanoregula*). The highest relative abundance of methanogens

(0.012 %) occurred within station PP07 (key site S1) (Fig. 6a), and the other key sites with

the highest CH$_4$ concentrations did not exhibit particularly elevated methanogens abundances.

Conversely, bacterial putative groups associated with methanotrophy/methylotrophy,

particularly OTU within the family *Methylophilaceae*, were detected at all sites and ranged

between 3.5 to 5.5 % relative abundance (Fig. 6b). Restricting our analysis to genera known

to be strict methanotrophs, we find sequences affiliated with *Methylobacter* that range from

0.01 – 0.3 % relative abundance, and only traces of *Ca.* Methanoperedens (Supplemental Fig.

S9). The relative abundances of these groups were approximated to pseudo-absolute

abundances using the quantitative qPCR results from each sample. Patterns in methanogen

abundances were consistent regardless of scale (Fig. 6c), while methano-/methylotrophs





exhibited higher abundances within stations PP10, PP11 (key site S2), and PP23-PP25 (incl.

key sites S4 and S5), and lower abundances within PP06, PP09, PP15, PP17, and PP20 (key

site S3) (Fig. 6d).

The correlations between the total absolute abundances of archaeal microbial communities

against the water properties at stations (Fig. 7) show a positive linear correlation between $T$

and the abundance of methanogens ($r^2$=0.35, $p$ = 0.05) and methano-/methylotrophs ($r^2$=0.43)

(Figs. 7a and 7b). A negative linear correlation against κ ($r^2$=0.31 for methanogens and

$r^2$=0.24 for methano-/methylotrophs) (Figs. 7c and 7d). The $p$CH$_4$ at stations is also positively

correlated to the abundance of methanogens ($r^2$=0.11) and methano-/methylotrophs ($r^2$=0.21)

(Figs. 7e and 7f).

The average DOC measured in all of the sampling stations was 9.0±1.0 mg L$^{-1}$. The largest

DOC values were measured at some of the sampling stations located on the east side of the

transect at the confluence between tributaries and the Kolyma main stem or closer to the river

bank. The highest measured DOC value was 11.9 mg L$^{-1}$ at station PP05, located at the

confluence between the Panteleikha-Ambolikha rivers and the Kolyma main stem, whereas

the lowest DOC value was measured at station PP25 near the river bank at Duvannyi Yar (7.5

mg L$^{-1}$) despite the large turbidity observed in this site. Other sampling stations located at

confluences or near banks reached values ≥ 9 mg L$^{-1}$, i.e., PP07 and PP10 close to the

riverbank, PP18 and PP19 located inside and outside Leonid's stream respectively, as well as

station PP11 at the confluence to the Maly and Bolschoi Anjui (M&B Anjui) tributaries

(Table S1 and Fig. S10). No significant correlation (p < 0.1) was found between $p$CH$_4$ and

DOC.

### 3.4    Surface CH$_4$ emissions at transects and polygonal surface area at the Kolyma River section

The average gas transfer velocity during the sampling period was calculated with a hydraulic

model ($k_{R12}$=0.5±0.02 m d$^{-1}$) and a wind speed parameterization ($k_{W14}$=0.4±0.3 m d$^{-1}$) are in

close agreement. Because the magnitude of the flux density of CH$_4$ calculated in both

transects with these two $k$ values does not differ considerably (i.e., $F_{R12}$=0.02±0.007 mmol m$^{-2}$ d$^{-1}$ and $F_{W14}$=0.01±0.01 mmol m$^{-2}$ d$^{-1}$), we chose to present only $F_{R12}$ calculated using $k_{R12}$

after the hydraulic model. $F_{R12}$ will be presented hereinafter to as the flux density of CH$_4$,

FCH$_4$.

The average FCH$_4$ of CH$_4$ along the UP transect was 0.019±0.005 mmol m$^{-2}$ d$^{-1}$ and along the

DOWN transect was 0.026±0.008 mmol m$^{-2}$ d$^{-1}$. Maximum FCH$_4$ values at key sites were

0.034 mmol m$^{-2}$ d$^{-1}$ for site S5 during the UP transect, and 0.045 mmol m$^{-2}$ d$^{-1}$ at the key site



S2 during the DOWN transect (Fig. 3). Average $FCH_4$ in both transects was 1.5 times higher at key sites than in the other sites of the transects (Fig. 7). This is relevant considering that the

surface area represented by the key sites is 8 to 12 times smaller than the rest of the transects (calculated considering the navigated distance times a radius of 50 m around the sampling point).

The cumulative sum of the $CH_4$ fluxes in the UP transect for the period of sampling (1 day) was 14.4 mmol $m^{-2}$. At key sites, the fluxes accounted for 13 % (1.8 mmol $m^{-2}$) of the total

flux. In the DOWN transect, the total CH4 flux was 10.5 mmol $m^{-2}$, and the contribution of the key sites to the total emissions increased to 20 % (2.1 mmol $m^{-2}$).

We also calculated $FCH_4$ for a smaller stream (Leonid's stream) and the Ambolikha River (second-order tributary of Kolyma River) (Fig. 1), that were navigated during the DOWN transect on 17 June 2019. These navigated sections were not included in our estimate for main

channel. The average $FCH_4$ at the Ambolikha River (0.17±0.008 mmol $m^{-2}$ $d^{-1}$) and at the Leonid's stream (0.05±0.02 mmol $m^{-2}$ $d^{-1}$) were nearly five and two times higher respectively than at the key sites of the main channel during the DOWN transect (Fig. 7).

Based on the modeled $pCH_4$ in the gridded surface area of the Kolyma River section, we calculated the corresponding $FCH_4$ that would have been emitted through the total surface of

the river section (236.3 $km^2$). The total $CH_4$ flux at the surface of the river section during the UP transect is calculated as 934.2 mmol $d^{-1}$ (or $1.1 \times 10^4$ mgC $d^{-1}$), and for the DOWN transect is 1391.9 mmol $d^{-1}$ (or $1.7 \times 10^{14}$ mgC $d^{-1}$). This estimation allows for the calculation of the flux of gas through the entire surface area of the river section (and not only at the transect locations). We estimated an average of $3.3 \times 10^{12}$ mgC $d^{-1}$ (equivalent to 3300 tC $d^{-1}$) of the

total $CH_4$ emitted through the surface of the Kolyma River section during the sampling time of both transects (15-17 June 2019).

## 4    Discussion

### 4.1    Patterns and indicators of the spatial distribution of methane in Kolyma River and associated tributaries and streams

In June 2019, the Kolyma River exhibited large $pCH_4$ values that were up to 1,300 % supersaturated (equivalent to 28.3±8.5 µatm) with respect to atmospheric equilibrium. These values are comparable to measurements reported for summer in the main channel of the Lena River, i.e., 18 to 51 µatm, calculated from 30 to 85 nmol $L^{-1}$ for a $T$=14 °C in freshwater; (Bussmann, 2013). However, a large range in $pCH_4$ values has been measured in other Arctic

Rivers, such that the average $pCH_4$ in the Kolyma River is three times higher than measurements at the main channel of the Yukon River in North America (8.4 µatm) (Striegl



et al., 2012), and almost nine times lower than the mean $p$CH$_4$ value (236 µatm) in surface
waters of Kuparuk River in Alaska (Kling et al., 1992).

Our highly spatially-resolved underway continuous measurements of surface dissolved CH$_4$
were pivotal to reveal spatial variabilities and features in the main river channel that cannot be
obtained with sparse discrete sampling. The surface distribution of $p$CH$_4$ measured in a ~120
km section of Kolyma River was heterogeneous, with nearly two-fold higher concentrations
observed along riverbanks and near the confluence of tributaries (69 nmol L$^{-1}$, or $p$CH$_4$=41.1
µatm) than at the central parts of the river (46 nmol L$^{-1}$, or $p$CH$_4$=27.8 µatm) (Fig. 2 and
Table 1). Previous studies have demonstrated the influence of land to the distribution of
riverine CH$_4$ concentrations, for example along the Danube River (Canning et al., 2021b), and
within the Lena River (Bussmann, 2013). The concentration of dissolved CH$_4$ in Arctic sites
with direct contact to adjacent lands, such as in small tributaries, streams, lakes channels or
ponds, has been shown to be two to five times higher than what is observed in the main stems
of large rivers (Bussmann, 2013; Dean et al., 2020; Kling et al., 1992; Striegl et al., 2012). In
samples from creeks draining from permafrost into the Lena River, CH$_4$ concentrations (1505
nmol L$^{-1}$, or $p$CH$_4$ of 900 µatm) were between twenty to fifty times higher than in fluvial
waters (Bussmann, 2013). At the Lena Delta, the concentrations of CH$_4$ are higher (212 nmol
L$^{-1}$, or $p$CH$_4$ of 114.7 µatm, $T$=9.8°C and $S$=2.45), because they were directly influenced by
bottom soils (Bussmann et al., 2017). In tributaries of the Yukon River, the CH$_4$
concentrations were up to 690 nmol L$^{-1}$, being two times higher than in the main stem of the
same river (290 nmol L$^{-1}$) (Striegl et al., 2012). Similarly, our results show that, besides the
in-stream variability, tributary or stream CH$_4$ concentrations measured at the Ambolikha
River and Leonid's stream, were between two to six times higher than those in the main
channel of Kolyma River.

The average $p$CH$_4$ measured at the Ambolikha River (206.8±9.8 µatm) is consistent with the
measurements at the Kuparuk River (236 µatm) (Striegl et al., 2012), and the mean $p$CH$_4$
(292±109 µatm) measured during a 38-day time-series study that started 9 days after the
present study (i.e., on 26 June 2019) at a site in the Ambolikha River (Castro-Morales et al.,
2022). Whereas the average CH$_4$ concentration measured at Leonid's stream was 67 µatm
(111 nM), which is in the same order of magnitude as the maximum value measured at the
plume of Kolyma River at the East Siberian Arctic shelf in the summer of 2004 (55 µatm,
obtained from the reported 110 nM, $T$=5 °C and $S$=14) (Shakhova and Semiletov, 2007).

We characterized the spatial distribution of riverine $p$CH$_4$ as a function of temperature ($T$),
specific conductivity ($\kappa$) and the distance from the river banks ($z$), as suitable indicators for



the distribution of $CH_4$ during the late spring over larger areas of the Kolyma River (and potentially applicable to other Arctic rivers). We found that the distance to river banks is an indicator of the proximity to potential terrestrial $CH_4$ sources, hence it can be a useful benchmark for understanding the distribution and fate of $CH_4$ in natural surface waters (Fig.
4). With a statistical approach, we used the selected predictors to fill gaps in areas of the river where no $CH_4$ data were available (Fig. S6). Similar approaches could be used to improve the $CH_4$ data currently available for the global $CH_4$ budget (Saunois et al., 2020) and to aid in forecasting riverine $CH_4$ following the projected increases in warmer river waters, abrupt permafrost thawing, and collapse of riverbanks.

## 4.2    Identification of microbial communities associated with the riverine $CH_4$ concentrations

Overall patterns in microbial community composition, e.g., the similarities in the relative abundances of bacterial and archaeal groups, were also strongly related to the temperature and specific conductivity of the river water (Fig. 5). Unlike with $CH_4$, distance to shore was not
apparent in explaining differences in community composition. Arctic riverine microbial communities track closely with water temperature, flow rate, and biogeochemistry (Campeau and del Giorgio, 2014; Crump et al., 2009) and match patterns in DOM composition and concentration (Castro-Morales et al., 2022; Kaiser et al., 2017). The strong explanatory power of temperature and specific conductivity we observe in this study fits in with the concept of
riverine community coalescence as they approximate the mixing of distinct water sources over a spatially small region, whereby the dynamic community assemblage mechanisms are inextricably linked to transport processes and rapidly changing selective pressures (Mansour et al., 2018). In this sense, spatial patterns in community composition can act as robust bioindicators of the relative inputs of transported metabolic end products derived from
terrestrial sources, like $CH_4$ or $CO_2$. To support the relationship between community composition and the originating source of $CH_4$, we examined the distributions of functional microbial groups putatively associated with $CH_4$ production and consumption. The strongest evidence was the overlap in detected methanotrophs and methanogens within our study and a previous study by Kwon et al. (2017), that examined these groups within permafrost soils
adjacent to our site (PP09). More specifically the highest relative abundances of groups associated with *Methanobacterium* and *Methylobacter* in both the surficial soils and our discrete water samples.

Expanding on this, biological $CH_4$ production is only known from the Archaeal domain of life, and methanogens, as strict anaerobes (Evans et al., 2019), are unsuited to grow within



oxic river waters. We suggest that the relative and pseudo-absolute abundances of sequences
affiliated with methanogens further act as more specific indicators of sources originating from
anoxic, terrestrial $CH_4$ hotspots, as supported by the statistically significant correlation
between methanogen abundance and methane concentration (Fig. 7). Additionally, we
anticipate that the methanogenic archaea exhibit longer residence times than $CH_4$ itself due to

its high diffusion and oxidation rates. The presence of soil-derived methanogens in the river
water might be indicative of even higher riverine $CH_4$ concentrations, as part of it can be
already outgassed or oxidized. The weaker correlation of methane to methanogen abundance
compared to temperature or specific conductivity, parameters expected to change slower than
methane concentrations, likely reflects these differences in transport mechanisms. Of the two

methanogens we detected, *Methanobacterium* was recently shown to be the primary
methanogen detected in the surface waters of thermokarst ponds and is more typical of acidic
and peat-dominated aquatic ecosystems (Vigneron et al., 2019). *Methanoregula* (within order
*Methanomicrobiales*) have also been shown to be abundant groups within permafrost thaw
lakes (Crevecoeur et al., 2016) and were suggested to be more typical of deeper and less

acidic water bodies (Vigneron et al., 2019) (Fig. 6).
Conversely, we expected microbial groups that consume $CH_4$ to also be indicative of $CH_4$
sources into the river. Groups affiliated with methylotrophy (e.g., *Methylophilaceae* –
*Methylotenera*) exhibited ten times higher relative abundances than groups of strict
methanotrophic organisms (*Methylobacter*) (Fig. 6), suggesting that in addition to methane,

other sources like methanol associated to the degradation of $CO_2$ by methanotrophs (Xin et
al., 2007) or by some groups of phytoplankton (Mincer and Aicher, 2016), were sources of
carbon in this environment. In support of this finding, aerobic methanotrophs have been found
at much higher relative abundances (>25 %) and higher diversity within thermokarst well-
stratified subarctic Canadian ponds, than the maximum of 0.3 % detected here, where distinct

genera (Methylobacter and Methylomonas) within the order *Methylococcales* where the most
abundant (Crevecoeur et al., 2015; Vigneron et al., 2019). This is a sensical finding, as the
dynamic river flow enables the diffusive methane transport and emissions to the atmosphere
compared to the emissions across smaller surface areas in highly-stratified, less dynamic and
largely anoxic pond environments. The majority of the $CH_4$ produced in thawing permafrost

is first locally oxidized before it can be released to the atmosphere (Olid et al., 2021). Thus,
the higher relative abundance of $CH_4$-consuming bacteria compared to $CH_4$-producing
archaea in the Kolyma River suggests that a considerable fraction of $CH_4$ is already oxidized
within the recently thawed active layer.





### 4.3 Temporal variability of methane in Kolyma River

Our continuous high-resolution measurements of $p$CH$_4$ in the Kolyma River allowed us also to identify a large temporal variability in spite of the short time scale of our measurements. The differences in the $p$CH$_4$ and FCH$_4$ (flux density of methane) between the UP and DOWN transects might be due to a rapid response to changes in CH$_4$ driven by the interactions between the main flow of the river and the continuous contribution of external CH$_4$ inputs

resulting from melting, rather than by an advective signal travelling down the main channel of Kolyma River. Still, our measurements cannot represent any mid- to long-term CH$_4$ variation in the river, and the differences between the transects might also be due to different spatial locations.

The Kolyma River Basin is the only one in the Arctic completely underlain by continuous

permafrost, which could result in even higher soil CH$_4$ production and release into the river network during permafrost thaw compared to other Arctic rivers. During the Arctic melt season (May to June), the surface hydrologic connectivity between the land and rivers is enhanced. As the seasonal progression takes place, deeper water-saturated soil layers are thawed, and substances, microorganisms, and gases, like CH$_4$, are mobilized through the

lateral transfer from groundwater discharge into Arctic inland waters, particularly to the fluvial network (Connolly et al., 2020; Harms et al., 2020; Saunois et al., 2020). It has been demonstrated that the majority of the CH$_4$ emitted to the atmosphere from subarctic ponds is sustained by the discharge of CH$_4$ from groundwaters upon the active layer thaw (Olid et al., 2021).

### 4.4 Methane emissions in Kolyma River and comparison to other estimates

The average estimated annual flux in the polygon section at the Kolyma River during our sampling is $0.63 \times 10^{12}$ g CH$_4$ yr$^{-1}$ (or 0.63 Tg CH$_4$ yr$^{-1}$, for a 146-days ice-free season between 20$^{th}$ May and 12$^{th}$ October 2019 obtained from the river discharge curve, Fig. S1) considering a surface area of 236.3 km$^2$. These emissions are on the same order of magnitude

as the annual flux of CH$_4$ at the East Siberian Arctic Shelf (ESAS) estimated to be $0.11 \times 10^{12}$ g CH$_4$ yr$^{-1}$ (or 0.11 Tg CH$_4$ yr$^{-1}$, for a 90-days ice-free season) in summer of 2003 and 2004 for a surface area of $1.0 \times 10^6$ km$^2$ (which is orders of magnitude greater than the polygon section of Kolyma River) (Shakhova and Semiletov, 2007). In Arctic shelves, the concentration of CH$_4$ is strongly influenced by riverine inputs, particularly in to bottom layers

of shelf waters due to differential water density gradients (Shakhova and Semiletov, 2007). Decreasing flow velocities (i.e., discharge) allow sedimentation of organic matter in the delta areas, stimulating microbial sedimentary processes that finally lead to the formation of CH$_4$



and $CO_2$. Dropping water levels during summer also facilitates $CH_4$ emissions from riverine sediments to the atmosphere. This has been observed in the Lena River region, where

contributions from bottom surface sediments are more significant to the measured $CH_4$ concentrations than riverine lateral exports (Bussmann et al., 2017).

Taking into account a polygon surface area of 236.3 $km^2$ and 146 days of ice-free water, we estimated a total flux of 0.63 Tg $CH_4$ $yr^{-1}$ in the investigated river section. This calculation is by far robust and largely uncertain, considering that our measurements only correspond to a

short-term data set during the open water season, and that large temporal and spatial variations in relation to e.g., changes in water sources, temperature regime and lateral carbon inputs throughout the ice-free period is expected. This has been recently demonstrated at the Ambolikha River (tributary of the Kolyma River), where riverine $CH_4$ concentrations decreased over time during the open water season due to persistent emissions to the

atmosphere dominating over declining external gas inputs during the summer low flow (Castro-Morales et al., 2022). Still, the annual $CH_4$ flux value provided here for the investigated Kolyma River section, provides an upper end of the potential magnitude and relevance of $CH_4$ atmospheric emissions from an Arctic River.

As both the oxidation rates and the diffusive emissions of $CH_4$ through the water-atmosphere

interface are faster processes than the lateral gas transport in the water column. Thus, despite the large $CH_4$ concentrations and emissions identified in the upstream river waters, the surface riverine $CH_4$ measured >100 km upstream of the shelf is locally emitted (or oxidized) and does not influence the surface $CH_4$ concentrations measured at the river plume and at the East Siberian Arctic Shelf.

Morphology and stream size seem to be also key parameters for the amount of gas delivered from land and emitted through the water surface into the atmosphere, as the potential for large gas emissions is higher in smaller streams with shorter water travel distances. Our data support this assumption, as the $FCH_4$ at key sites was two to five times lower than the average $FCH_4$ at the smaller Leonid's stream and Ambolikha River respectively (Fig. 8). The surface

areas of the key sites characterized by elevated $FCH_4$ are between 8 to 12 times smaller than the surface area covered by the rest of the transect. However, the $CH_4$ emissions at key sites were 1.5 times higher than in the other sites, and represent between 13 to 20 % of the total cumulative emissions in both transects.

Because the diffusion of $CH_4$ in water is slower than in air, riverbanks can thus act as efficient

vectors for the local emissions of $CH_4$ formed and stored in the subsoil. The projected increase in freshwater inputs, deepening of active layers, and increase in soil drainage, as



more permafrost is thawing in response to warmer and wetter Arctic summers (AMAP, 2017;
Bring et al., 2016; Bussmann et al., 2017; Chiasson-Poirier et al., 2020), will enhance the
input of $CH_4$ from external terrestrial sources at hotspots over extended periods during the
open water season. Additionally, projected longer ice-free periods in the Arctic, i.e., an earlier
start of melt periods and longer open water seasons, can therefore lead to an increase in $CH_4$
emissions from inland waters (Wik et al., 2016). This ultimately will have an impact on the
current $CH_4$ budget of the Arctic. By not considering the variable aquatic ecosystems and
water cycle of the Arctic, the estimated 4 to 5 % contribution of high latitudes to the total
global methane emissions (Rosentreter et al., 2021; Saunois et al., 2020) may be
underestimated.

The irregular location of $CH_4$ hot spots along the river banks and their potentially continuous
elevated $CH_4$ contributions to the river, possess a challenge to estimating lateral transport of
$CH_4$ from upstream to downstream waters. Elevated $CH_4$ concentrations at the Arctic shelves
are thus primarily influenced by local sources (i.e., bottom soils and degrading shelves)
(Shakhova and Semiletov, 2007). However, to improve the estimates of riverine $CH_4$
concentrations that can actually reach the ocean in the context of increasing warming and
thawing, and to improve the knowledge of the contribution of Arctic rivers and streams to the
regional and global $CH_4$ budgets, it is necessary to intensify the spatial and temporal
resolution of the direct measurements of $CH_4$ in Arctic Rivers.

## 5    Conclusions

In this study, we measured for the first time continuous high-resolution $pCH_4$ in a large
section of Kolyma River during the late freshet of 2019, and combined these observations
with microbial community analysis in water samples to investigate the potential source of this
gas. The large spatial variability of surface methane concentrations in the river channel was
associated with hotspots located at the river bank and at confluences with tributaries where
methane was almost two times higher than at the center of the channel. The identified
presence of methane-producing archaea in a well oxygenated river water suggests that most of
the $CH_4$ is laterally transported from external terrestrial sources into the river channel, rather
than produced within the river water. Elevated riverine local methane emissions were
associated with identified hotspot areas on land suggesting efficient linkages between the land
and the aquatic ecosystems. Upstream river boundaries do not seem to be a source of $CH_4$ into
the Arctic Ocean via downstream transport with the river flow. Without continuous
measurements, it will remain unclear how much $CH_4$ is actually transported and emitted at the
peak of the melt period at the highest annual river discharge. Certainly, more abrupt collapses,





erosion and thawing of the Arctic Ocean shelves may contribute to the liberation and transport of soil-derived $CH_4$ into the ocean, as well as subsequent emissions into the atmosphere. Our results provide a glimpse of the potential contribution of methane emissions from Arctic Rivers, adding up to the largely unknown contributions from permafrost and inland waters.


**Data availability**

The data presented in this work will be made available through a link to the Zenodo public repository upon the publication of this work.

**Competing interests**

The authors declare that they have no conflict of interest

**Author contribution**

K.C.-M. conceived and designed the study. K.C.-M, A.C., A.K., K.K., S.A., O.K. and N.Z.

contributed to fieldwork and logistics. K.C.-M, A.C., M.G., S.A., W.A.O. contributed to lab

work, sample and data analyses. K.C.-M., A.C. and W.A.O wrote the paper with contributions

from all authors in draft versions prior to submission.

**Acknowledgments**

This work was conceived within the project PROPERAQUA funded by the Deutsche Forschungsgemeinschaft (DFG, German Research Foundation) (project No. 396657413). The contributions from AC and AK were funded by the MOSES program of the Helmholtz Association and the C-CASCADES ITN of the EU (project No. 643052). KK and WAO were supported by the Collaborative Research Centre 1076 AquaDiva (CRC AquaDiva) funded by

DFG (project No. ID 218627073). MG and OK were supported through funding by the European Commission (INTAROS project, H2020-BG-09-2016, Grant Agreement No. 727890, Nunataryuk project, H2020-BG-11-2016/17, Grant Agreement No. 773421). Especial thanks to the Northeast Scientific Station and the Pleistocene Park in Chersky for their invaluable assistance during fieldwork. Thanks to Dr. Robert Lehmann for the analysis

of DOC samples at the FSU-Jena.



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



**Figures**

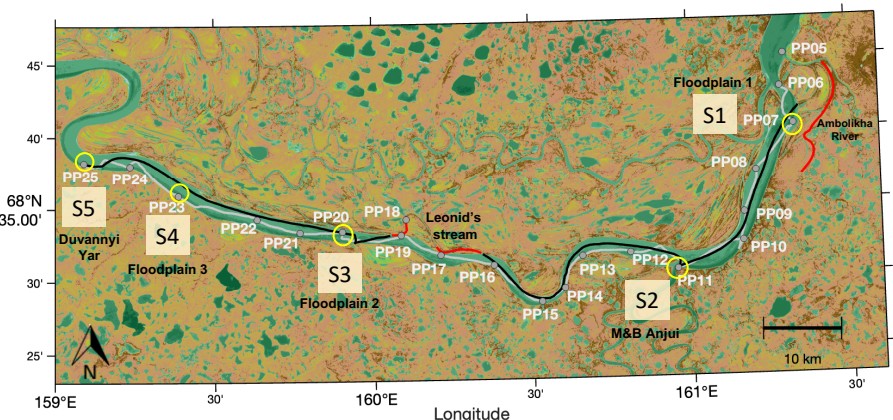

**Figure 1 –** Navigated transects in the Kolyma River: upstream (UP) (grey line, sampled from 15 June 2019 at 12:30 h to 16 June 2019 at 16:59 h) and downstream (DOWN) (black line, navigated on 16-17 June 2019). Gaps in the continuous UP and DOWN transects are data not considered for the analysis because they involved navigation outside the main river channel (i.e., transects at Leonid's stream and the Ambolikha River indicated in red). Discrete samples

were collected in 21 sampling stations (PP05-PP25) during the UP transect (grey markers). Key sites (and stations): S1 (PP07), S2 (PP11), S3 (PP20), S4 (PP23), and S5 (PP25) are circled in yellow. This map was created using MATLAB® with data from a composite image for June, July and August from 205-2018 using Sentinel-2 NDVI maps (https://developers.google.com/earth-engine/datasets/catalog/sentinel).







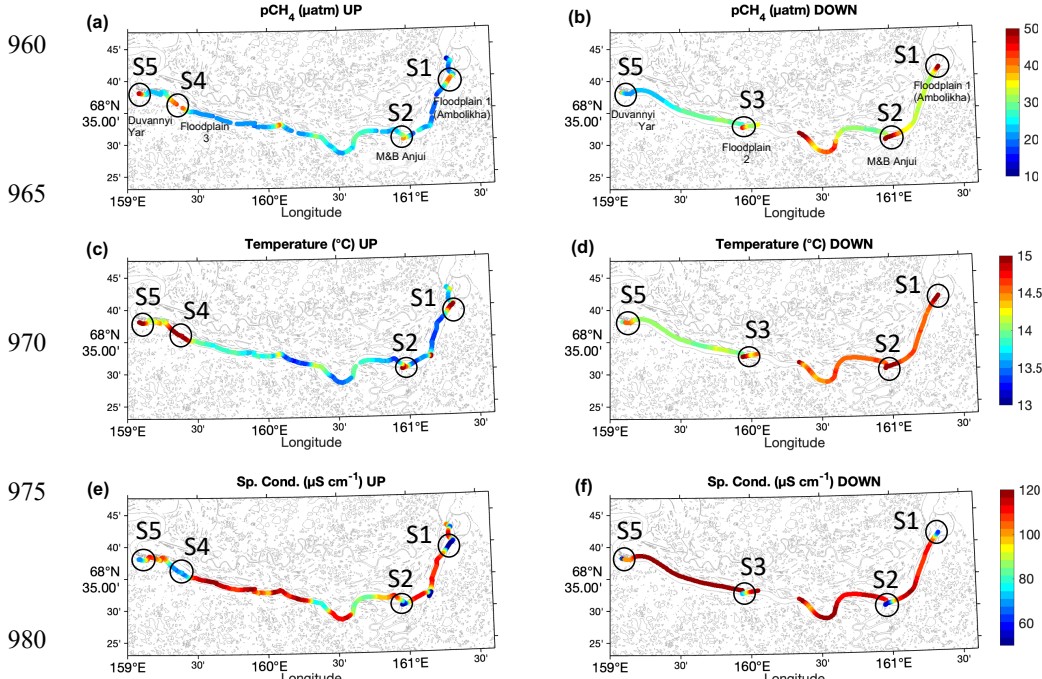

**Figure 2** – Spatial distribution of water properties measured along transects UP (left) and DOWN (right) at the main stem of the Kolyma River for $p$CH$_4$ (a and b), $T$ (c and d), and $\kappa$ (e and f). The location of key sites S1 to S5 are indicated. The values corresponding to Ambolikha River and Leonid's stream are not shown.



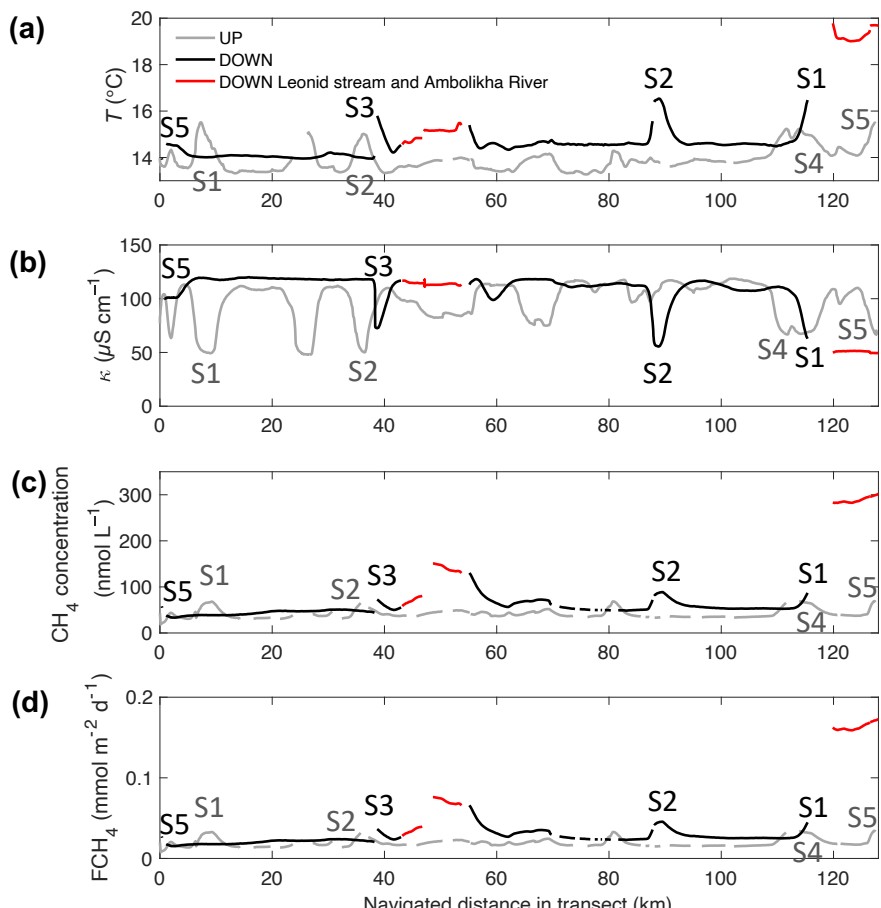

Water properties measured along the UP and DOWN transect in Kolyma River (15-17 June 2019)

**Figure 3** – Water properties measured in transects UP (grey) and DOWN (black): a) water temperature, $T$; b) water-specific conductivity, $\kappa$; c) $CH_4$ concentration, $C_w$, and d) flux density of $CH_4$, $FCH_4$, all shown as a function of the navigated distance (km) along each transect. The location corresponding to the key sites S1 to S5 are indicated and color-coded in each signal (light grey – UP transect and black – DOWN transect). The Ambolikha River and Leonid's stream are shown in red. Gaps in the data indicate erroneous or not measured data in the transect.





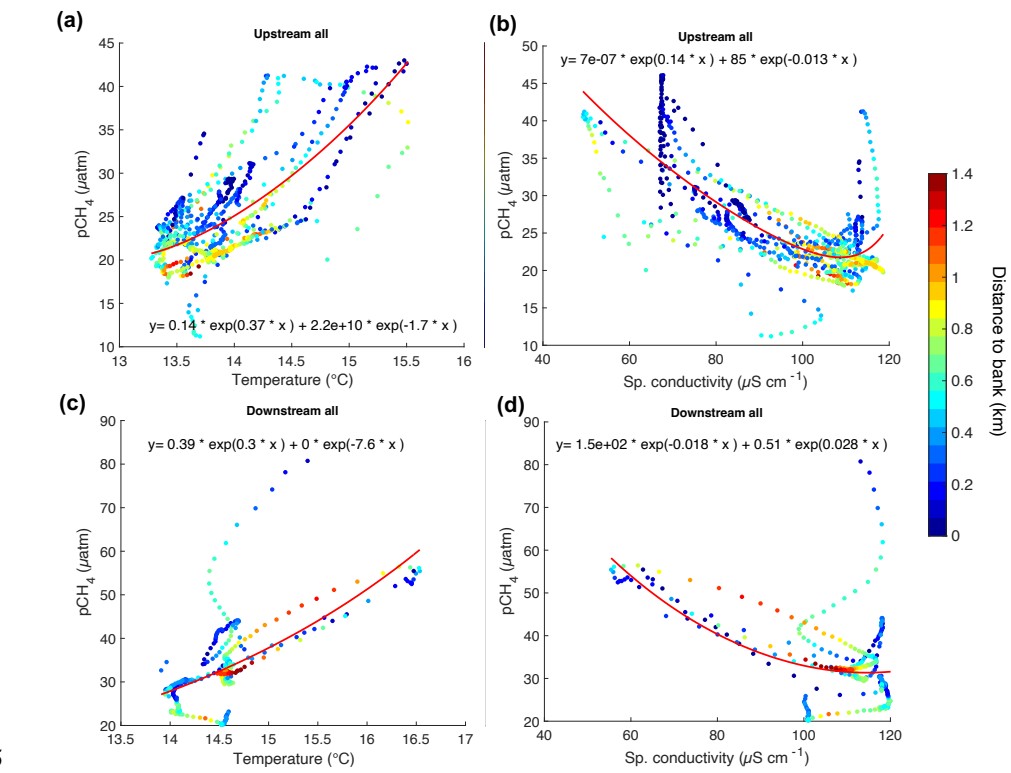


**Figure 4** – Correlation graphs for UP (a and b) and DOWN (c and d) transects between $T$, $\kappa$
and $p\mathrm{CH}_4$ as a function of the distance to bank ($z$ in km) indicated in the color scale.










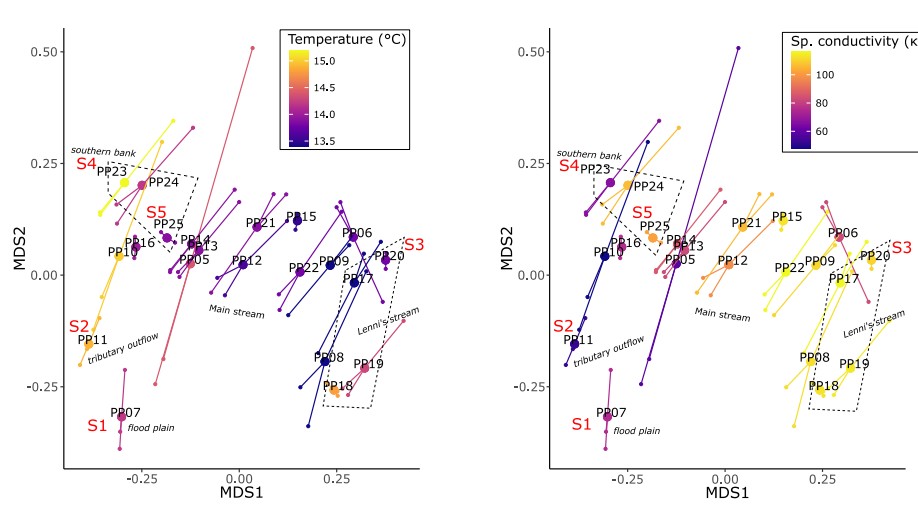

**Figure 5** – Riverine microbial community composition linked to temperature (left) and specific conductivity (right). Both plots represent the same underlying community data, with dissimilarities determined by the Bray-Curtis metric and visualized with non-metric multidimensional scaling plots.










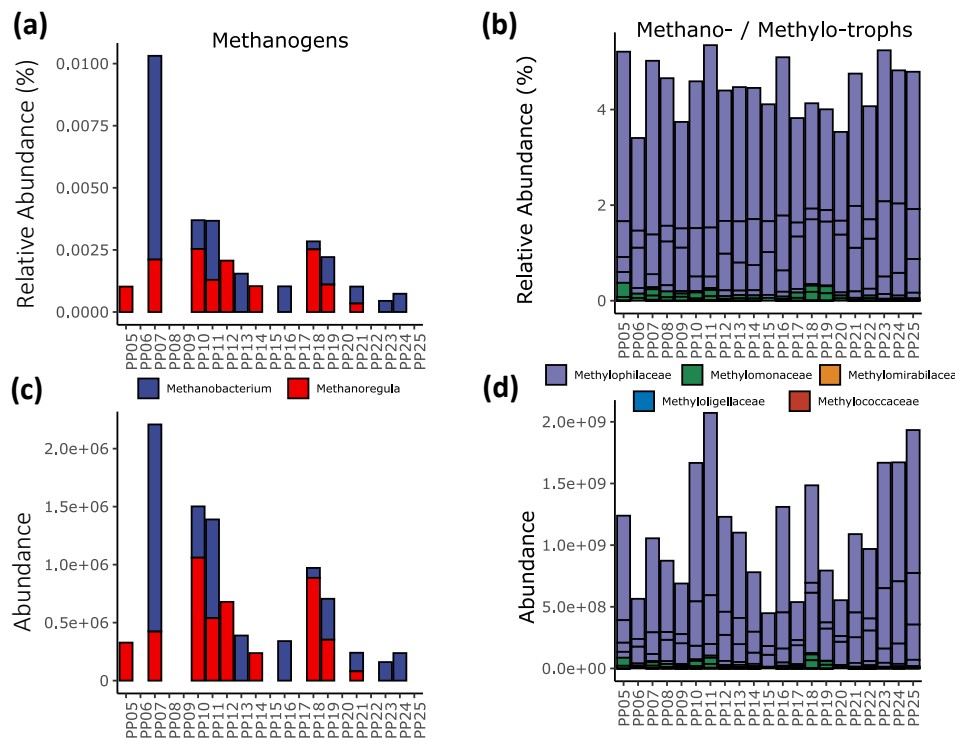

**Figure 6** – Relative (top) and pseudo-absolute (bottom) abundances of putatively
methanogenic archaeal genera (left) and methylotrophic bacterial families (right). An
expanded version that includes only the methanotrophs is available in the supplemental
information.





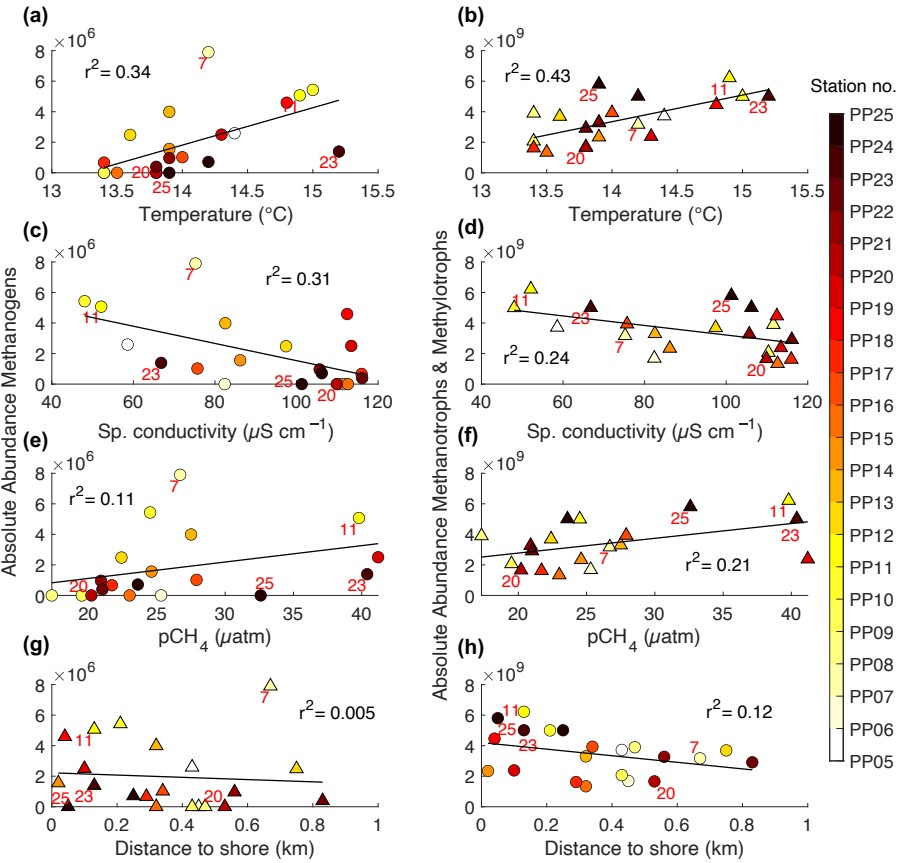


**Figure 7** – Linear correlations between the total absolute abundances of archaeal microbial communities (left, methanogens and right, methanotrophs) and the 1-min averages of water properties measured at the 21 sampling stations along the DOWN transect in Kolyma River. Red numbers in some of the markers indicate the station number corresponding to the key
sites S1 to S5.





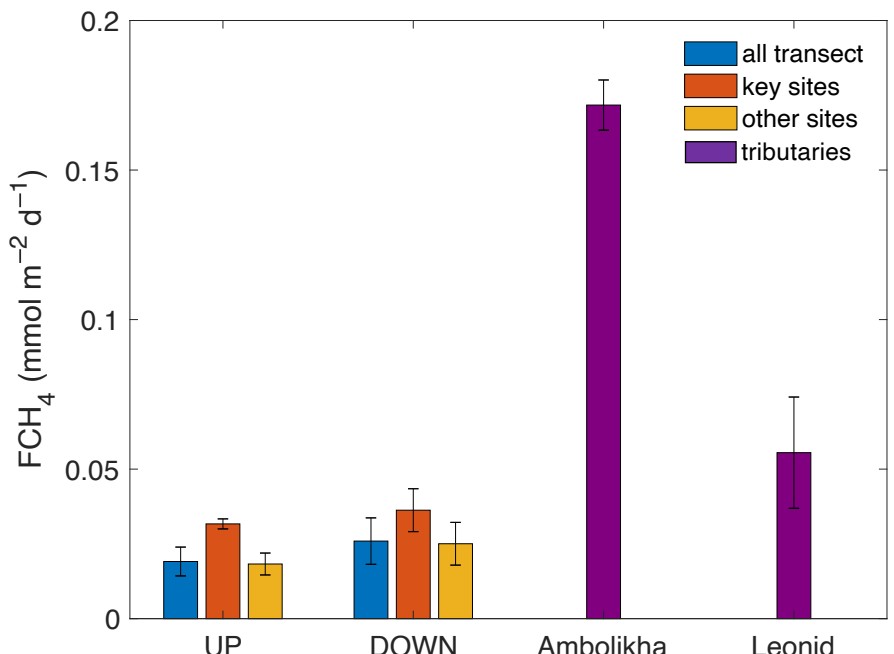


**Figure 8** – Average flux density of $CH_4$ ($FCH_4$) calculated for the entire UP and DOWN transects, and for the key sites and other sites. $FCH_4$ for the tributaries Ambolikha River and Leonid's stream are also shown. Error bars denote the standard deviation of the mean.






**Table 1 –** Average ± 1 std. deviation (minimum and maximum values below it) of $p$CH$_4$, the concentration of CH$_4$ ($C_w$), $T$ and $\kappa$ measured along the UP and DOWN transects in key sites (S1 – S5) and in the other sites of each transect. Measurements done in a tributary (Ambolikha River) and a stream (Leonid's stream) as part of the measurements during the DOWN transect are also shown.


| Location | $p$CH$_4$ (µatm) | $C_w$ (nmol L$^{-1}$) | $T$ (°C) | $\kappa$ (µS cm$^{-1}$) |
|---|---|---|---|---|
| Both transects | 28.3±8.5 (11.2 – 80.7) | 45.9±12.9 (18.9 – 130.2) | 14.1±0.6 | 96.8±21.5 |
| UP transect | 25.8±6.7 (11.2 – 46.1) | 41.5±9.2 (18.9 – 69.2) | 13.9±0.6 | 92.2±22.2 |
| UP key sites | 39.4±4.3 | 65.0±3.0 | 14.9±0.3 | 65.1±6.9 |
| UP other sites | 23.8±4.3 | 39.9±7.0 | 13.9±0.5 | 95.8±20.9 |
| DOWN transect | 33.2±9.4 (20.2 – 80.7) | 54.3±14.7 (33.3 – 130.2) | 14.5±0.5 | 108.3±14.5 |
| DOWN key sites | 42.8±9.2 | 72.4±12.4 | 15.7±0.8 | 76.0±16.0 |
| DOWN other sites | 31.8±8.5 | 52.7±13.7 | 14.4±0.3 | 112.4±7.3 |
| Ambolikha River (DOWN) | 206.8±9.8 (191.7 – 222.9) | 300.7±12.1 (282.2 – 320.7) | 19.6±0.3 | 49.9±0.9 |
| Leonid's stream (DOWN) | 66.8±22.0 (37.0 – 92.9) | 111.1±35.7 (60.8 – 150.7) | 15.1±0.3 | 113.9±1.6 |