# Peer review of "Highest methane concentrations in an Arctic River linked to local terrestrial inputs."

_Biogeosciences, 2022_

## Author Comment (AC2)

**Author comments for manuscript bg-2022-135**

*„The highest methane concentrations in an Arctic river are linked to local terrestrial inputs"*
Karel Castro-Morales, Anna Canning, Sophie Arzberger, Will A. Overholt, Kirsten Küsel, Olaf Kolle, Mathias Göckede, Nikita Zimov, and Arne Körtzinger

On behalf of all co-authors, we thank the support from the associate editor Alexey V. Eliseev and the useful comments from two anonymous reviewers for the improvement of our work. The answers to each of the referees' comments are provided below in blue font color.

**Referee #2**

Summary of the paper: *The highest methane concentrations in an Arctic river are linked to local terrestrial inputs*. Castro-Morales et al. studied methane concentrations on a 120 km portion of the Kolyma River during the late freshet (twice in June 2019). They observed a strong spatial disparity along the river bed with higher concentrations linked to warmer temperature, low conductivity and closeness to the river bank. This high spatial resolution study in the Kolyma River is the key start to better understand methane concentration pattern in (Arctic) rivers and their potential as methane source/sink. The correlation with temperature is interesting as it poses the question of increasing methane emission from Arctic rivers during the current global warming. It would be great to confirm if the microbes detected during this study are alive (active in the river itself) or dead (originating from the nearby permafrost surficial soils as stated by the authors) by sequencing RNA, although this type of sampling comes with logistical challenges in such remote environments. Overall this study is well done and brings key findings.

Thank you for your supportive comments. We do agree that measuring the activity of methanogens in the river water would be ideal to answer the question of the source of riverine $CH_4$. As the reviewer #2 pointed out, due to logistical constrains, we were not able to extract RNA from the samples for doing metatranscriptomics analyses.

Main comments:

- Why did the authors choose to only study dissolved organic carbon (DOC) as a food source of their methanogens while particulate organic carbon (POC) could a food source as well? Especially because the GF/F filter used to filter the water for DOC could be used to quantify POC. Eventually, DOC concentrations are not used at all in this study and could be removed.

The reviewer is correct that POC should have been included for this analysis, and referee #1 also made the same comment. We did not explicitly measure POC in the water samples but rather total organic carbon, TOC (i.e., unfiltered water aliquot). We will replace the results of DOC by those of TOC. For further details in this response, please see also our comments to reviewer #1.

- Were there any new OTU detected during the study? If so they should be deposited in GenBank.

All sequences have been deposited in GenBank with the reference BioProject No. PRJNA881395. The data is openly available. This information will be added in the revised manuscript in the data availability section.

Minor comments:

L364, 369: Could you add the instrumental error on the measurement of $pCH_4$?

The accuracy or measurement error of the CONTROS HydroC® TDLAS sensor for measuring $pCH_4$ is according to the manufacturer (-4H- JENA) in a range of ±2 µatm or 3 % of the reading (Canning et al., 2021a). This information will be added in the revised manuscript.

L437: Was the river anoxic where Methanobacterium, and Methanoregula were detected? If not, are they rather active or dead (originating from the nearby terrestrial environment)? Did you consider methane production within the river as this as been detected in other aquatic environments (see Bogard et al., 2014 (Oxic water column methanogenesis as a major component of aquatic CH4 fluxes, Nature).

Good point. Thanks for pointing to the Bogard et al., (2014) paper. The river water was oxic at all stations with an average $O_2$ saturation of 110 % (see figure added below), which should preclude methanogenesis in the river water, as it has traditionally been assumed to occur only in anoxic environments. Only based on our DNA analyses, we are not able to distinguish between active and dead cells.

There is indeed increasing evidence of $CH_4$ production in oxic marine and freshwater environments (Bogard et al., 2014; Grossart et al., 2011). Oversaturation of $CH_4$ in surface waters of lakes has been shown to result from two processes: $CH_4$ release from littoral sediments in combination with horizontal transport to the open water and *in situ* net production of $CH_4$ in oxic surface water. Their relative importance is under debate (Bogard et al., 2014; Encinas Fernández et al., 2016; Grossart et al., 2011; Peeters and Hofmann, 2021). As discussed in Bogard et al. (2014), the link between oxic *in situ* $CH_4$ production and algal dynamics would not explain the high spatial heterogeneous $CH_4$ concentrations in the Arctic River water with a maximum detected close to river banks and near tributaries during the late freshet. Release of $CH_4$-rich pore water and of soil-borne methanogens during permafrost melting and resuspension events, rather than *in situ* net production of $CH_4$ in oxic surface by active methanogens might be the main driver.

Similarly, $CH_4$ concentrations in shallow water zones of lakes can be explained by release of $CH_4$-rich pore water during resuspension events and by elevated anaerobic $CH_4$ production in warmer sediments at shallow water depths compared to deep water sediments (Peeters and Hofmann, 2021). We will discuss this in the revised version.

[Figure]

Fig. 1 – Oxygen saturation in the river water along the stations in Kolyma River.

L457: $r^2 = 0.11$ and 0.21 are rather low correlation coefficients, can you add p values? If p is not significant this needs to be indicated so as to not mislead the reader on the presence of correlation.

We have added the statistical significance of the mentioned correlations between water properties and the absolute abundances of methano-/methylotrophs at stations. The modified paragraph reads as follows:

"The correlations between the total absolute abundances of archaeal microbial communities against the water properties at stations (Fig. 7) show statistically significant ($p<0.05$) positive linear correlations between T and the abundance of methanogens ($r^2=0.35$, $p = 0.005$) and methano-/methylotrophs ($r^2=0.43$, $p=0.001$) (Figs. 7a and 7b). A statistically significant negative linear correlation was obtained against κ ($r^2=0.31$, $p=0.007$) for methanogens and for methano-/methylotrophs ($r^2=0.24$, $p=0.02$) (Figs. 7c and 7d). The pCH$_4$ at stations is also positively correlated to the abundance of methano-/methylotrophs ($r^2=0.22$, $p=0.04$) and methanogens ($r^2=0.11$, $p=0.15$) (Figs. 7e and 7f), however the latter is not statistically significant at $p<0.05$".

L459-470: The paragraph on DOC concentration comes a bit out of the blue after the microbial analysis, maybe add its separate paragraph? Especially because it is not used afterwards

As mentioned in the comment above, we will substitute DOC by TOC in the analysis of results. Given that there is no clear evidence that TOC (or DOC) played an important role as indicators for the CH4 distribution in the river, and that the TOC data is only available for 8 stations, this paragraph will be shortened and only the correlation results will be presented.

L585: Some methanogens have been detected in oxic environments before, see Bogard et al., 2014 (Oxic water column methanogenesis as a major component of aquatic CH4 fluxes, Nature).

Thanks again for this comment. We agree that it would strengthen the manuscript by adding in the revised manuscript a paragraph addressing the possibility of methanogenesis in oxic waters. Detailed information has been mentioned in the response above

Typographical corrections:

Since you choose to simplify methane as "$CH_4$" please to do homogenously in the manuscript

Thanks for this comment, we will replace in all instances "methane" by "$CH_4$".

**References cited in this comment**

Bogard, J. M., del Giorgio, P. A., Boutet, L., Garcia Chavez, M. C., Prairie, Y. T., Merante, A., and Derry, A. M.: Oxic water column methanogenesis as major component of aquatic $CH_4$ fluxes, Nature Communications, 5, 5250, 10.1038/ncomms6350, 2014.

Canning, A., Körtzinger, A., Fietzek, P., and Rehder, G.: Technical note: seamless gas measurements across Land-Ocean Aquatic Continuum - corrections and evaluation of sensor data for $CO_2$, $CH_4$ and $O_2$ from field deployments in contrasting environments, Biogeosciences, 18, 1351-1373, 10.5194/bg-18-1351-2021, 2021a.

Encinas Fernández, J., Peeters, F., and Hofmann, H.: On the methane paradox: transport from shallow water zones rather than in situ methanogenesis is the major source of $CH_4$ in the open surace water of lakes, Journal of Geophysical Research: Biogeosciences, 121, 2717-2726, 10.1002/2016JG003586, 2016.

Grossart, H.-P., Frindte, K., Dziallas, C., Eckert, W., and Tang, K. W.: Microbial methane production in oxygenated water column of an oligotrophic lake, Proceedings of the National Academy of Sciences of the United States of America, 108, 19657-19661, 10.1073/pnas.1110716108, 2011.

Peeters, F. and Hofmann, H.: Oxic methanogenesis is only a minor source of lake-wide diffusive $CH_4$ emissions from lakes, Nature Communications, 12, 1206, 10.1038/s41467-021-21215-2, 2021.

---

## Author Response (AR1)

**Author comments for manuscript bg-2022-135**

*„The highest methane concentrations in an Arctic river are linked to local terrestrial inputs"*
Karel Castro-Morales, Anna Canning, Sophie Arzberger, Will A. Overholt, Kirsten Küsel,
Olaf Kolle, Mathias Göckede, Nikita Zimov, and Arne Körtzinger

On behalf of all co-authors, we thank the support from the associate editor Alexey V. Eliseev
and the useful comments from two anonymous reviewers for the improvement of our work.
The answers to each of the referees' comments are provided below in blue font color.

As a main remark, we suggest a change in the title which is more semantically correct, the
new title will be:
**"Highest methane concentrations in an Arctic River linked to local terrestrial inputs."**

**Referee #1**

The article considers an extremely important phenomenon – the entry of methane into the
water mass of the Arctic river from its catchment area. It is shown that methane is
distributed inhomogeneously in the transverse direction, its highest content is
characteristic of coastal areas. This work opens up the direction of research necessary to
assess the flow of methane into the coastal zone of water bodies during the melting of
permafrost. A large number of studied indicators gives grounds to confirm the results
obtained. The conclusion seems to be important that a considered fraction of CH4 is
already oxidized within the recently thawed active layer.

We thank the reviewer for highlighting the scientific contribution of our manuscript.

From the comments on the work, the following should be mentioned. Dissolved organic
carbon is among the studied indicators. Why do the authors use this indicator, and not the
total organic carbon, which seems more correct, since organic suspension can be a carbon
source for methanogens?

The reviewer is correct that TOC should have been included for this analysis, and this is also
mentioned by referee #2. We measured total organic carbon, TOC, in unfiltered water
samples. Unfortunately, the TOC data is incomplete due to loss of samples. Results are only
available for 7 out 21 stations (33 %). Still, we provide the results of the correlation analysis
to integrate TOC as indicator in the revised manuscript.

The trend of TOC for the available samples is similar to that of DOC along the river section,
thus we do not foresee any major change in terms of the interpretation as an indicator for
methane distribution. As stated in the original manuscript, the correlation between $pCH_4$ and
DOC was positive, but weak and not statistically significant ($r^2=0.005$, $p<0.1$). A correlation
between TOC and $pCH_4$ (for the available results) is negative and higher ($r^2=0.41$), but it is
also not statistically significant at $p<0.1$.

Due to the lack of correlation between TOC and $pCH_4$, there is no further use of these data
throughout the manuscript (as was also the case for DOC). Hence, in the revised manuscript
section 3.3. of results replaced the mentions of DOC by TOC, and present the correlation
results of TOC vs. $pCH_4$ (instead of DOC vs. $pCH_4$). The supplementary figure S10
presenting the distribution of DOC along the stations has been also removed.

The new paragraph in section 3.3. reads now:

*"The average TOC measured in 7 out of the 21 sampling stations was 7.5±0.7 mg $L^{-1}$(Table S1). Since organic matter in suspension can be an important carbon source for methanogens, we correlated TOC v $pCH_4$. A negative but not significant correlation at $p < 0.1$ was found between $pCH_4$ and TOC."*

The second remark concerns the authors' conclusion that the upstream river sections are not a source of CH4 entering the Arctic Ocean by transferring downstream with river runoff. It is possible that this is the case in low-water phases, and during the period of maximum flood flow, the time of water reaching can be significantly reduced. The study of the length of the river section from which methane enters the mouth in various phases of the water regime was not included in the list of research tasks, but this idea seems very relevant for further work.

We thank the reviewer for this remark. In our view, some of the riverine $CH_4$ could be transported to the Arctic Ocean predominantly in downstream waters under high flow regimes, and in combination with periods when high gas transfer from land to the river takes place. This is assuming that most of the $CH_4$ that is present in the river is of terrestrial origin and not produced via oxic methanogenesis in the river water.

Considering a distance of ca. 100 km between the study section and the Arctic Ocean, and our measurements being done only during the late freshet period, it is not possible to assess if some of the $CH_4$ can reach the Arctic Ocean without being oxidized in the river course, or emitted to the atmosphere. Our results do not allow estimating a reach length to the Arctic Ocean. For future work, we would recommend to include a tracer study with isotopically marked $CH_4$ (e.g., $\delta^{13}CH_4$, as in Faber et al., 1996), and follow it from the source along the river. Additionally, vertical contributions of $CH_4$ such as those from anoxic bottom soils should be taken into account. In the revised manuscript, we added a brief paragraph discussing this point in the conclusion section:

*"Our analysis does not reveal the reach length of the $CH_4$ measured from our site to downstream waters. We suggest that the $CH_4$ measured in waters 100 km upstream the Arctic Ocean, might not reach shelf waters and instead is locally emitted to the atmosphere or oxidized in the river course. For this specific purpose, future works should include stable isotope studies to trace the sources and pathways of the $CH_4$ in the river water."*

Another remark concerns the reference to Figure 7. The authors write: statistically significant correlation between methanogen abundance and methane concentration (Fig. 7). Figure 7 in the appendix shows other data.

Thanks for your remark. As indicated in the manuscript, we referred in the mentioned paragraph (L587-588 of the original manuscript) to Figure 7 of the main text and not in the appendix. Perhaps this was overlooked by the referee.

In addition, we added a correction in the revised manuscript regarding the correlation between $pCH_4$ and methano-/methylotrophs that is statistically significant ($r^2=0.22$, $p=0.04$), but the correlation between methanogens and $pCH_4$ is not statistically significant ($r^2=0.11$, $p=0.15$) at $p<0.05$. This is corrected in the section 3.3 of results of the revised manuscript:

*"The pCH$_4$ at stations is also positively correlated and statistically significant (at p<0.05) to the abundance of methano-/methylotrophs (r$^2$=0.22, p=0.04), but is not statistically significant when correlated with methanogens (Figs. 7e and 7f)."*

Despite the above comments, the work seems to be very important and should certainly be published.

Thank you to the referee for the support for the publication of our manuscript.

**Author comments for manuscript bg-2022-135**

*„The highest methane concentrations in an Arctic river are linked to local terrestrial inputs"*
Karel Castro-Morales, Anna Canning, Sophie Arzberger, Will A. Overholt, Kirsten Küsel, Olaf Kolle, Mathias Göckede, Nikita Zimov, and Arne Körtzinger

On behalf of all co-authors, we thank the support from the associate editor Alexey V. Eliseev and the useful comments from two anonymous reviewers for the improvement of our work. The answers to each of the referees' comments are provided below in blue font color.

As a main remark, we suggest a change in the title which is more semantically correct, the new title will be:
**"Highest methane concentrations in an Arctic River linked to local terrestrial inputs."**

**Referee #2**

Summary of the paper: *The highest methane concentrations in an Arctic river are linked to local terrestrial inputs*. Castro-Morales et al. studied methane concentrations on a 120 km portion of the Kolyma River during the late freshet (twice in June 2019). They observed a strong spatial disparity along the river bed with higher concentrations linked to warmer temperature, low conductivity and closeness to the river bank. This high spatial resolution study in the Kolyma River is the key start to better understand methane concentration pattern in (Arctic) rivers and their potential as methane source/sink. The correlation with temperature is interesting as it poses the question of increasing methane emission from Arctic rivers during the current global warming. It would be great to confirm if the microbes detected during this study are alive (active in the river itself) or dead (originating from the nearby permafrost surficial soils as stated by the authors) by sequencing RNA, although this type of sampling comes with logistical challenges in such remote environments. Overall this study is well done and brings key findings.

Thank you for your supportive comments. We do agree that measuring the activity of methanogens in the river water would be ideal to answer the question of the source of riverine $CH_4$. As the reviewer #2 pointed out, due to logistical constrains, we were not able to extract RNA from the samples for doing metatranscriptomics analyses.

Main comments:

- Why did the authors choose to only study dissolved organic carbon (DOC) as a food source of their methanogens while particulate organic carbon (POC) could a food source as well? Especially because the GF/F filter used to filter the water for DOC could be used to quantify POC. Eventually, DOC concentrations are not used at all in this study and could be removed.

The reviewer is correct that POC should have been included for this analysis, and referee #1 also made the same comment. We did not explicitly measure POC in the water samples but rather total organic carbon, TOC (i.e., unfiltered water aliquot). In the revised manuscript we replaced the results of DOC by those of TOC. For further details in this response, please see also our comments to reviewer #1.

The new paragraph with changes from DOC to TOC in section 3.3. of the revised manuscript reads now:

*"The average TOC measured in 7 out of the 21 sampling stations was 7.5±0.7 mg L$^{-1}$(Table S1). Since organic matter in suspension can be an important carbon source for methanogens, we correlated TOC v pCH$_4$. A negative but not significant correlation at p < 0.1 was found between pCH$_4$ and TOC."*

- Were there any new OTU detected during the study? If so they should be deposited in GenBank.

All sequences have been deposited in GenBank with the reference BioProject No. PRJNA881395. The data is openly available. This information has been added in the revised manuscript in the data availability section.

Minor comments:

L364, 369: Could you add the instrumental error on the measurement of pCH$_4$?

The accuracy or measurement error of the CONTROS HydroC® TDLAS sensor for measuring pCH$_4$ is in a range of ±2 µatm or 3 % of the reading (Canning et al., 2021a) according to the manufacturer standard specifications (-4H- JENA engineering GmbH, Jena, Germany). This information has been added in the revised manuscript.

L437: Was the river anoxic where Methanobacterium, and Methanoregula were detected? If not, are they rather active or dead (originating from the nearby terrestrial environment)? Did you consider methane production within the river as this as been detected in other aquatic environments (see Bogard et al., 2014 (Oxic water column methanogenesis as a major component of aquatic CH4 fluxes, Nature).

Good point. Thanks for pointing to the Bogard et al., (2014) paper. The river water was oxic at all stations with an average O$_2$ saturation of 110 % (see figure added below), which should preclude methanogenesis in the river water, as it has traditionally been assumed to occur only in anoxic environments.

[Figure]

Fig. 1 – Oxygen saturation in the river water along the stations in Kolyma River.

Only based on our DNA analyses, we are not able to distinguish between active and dead cells. To expand in this explanation, we have added the following paragraph in sectin 4.2 of the revised manuscript:

*"Our data shows that river water was oxic at all stations with an average $O_2$ saturation of 110 %, which should preclude methanogenesis. However, there is increasing evidence that there is $CH_4$ production in oxic marine and freshwaters, and a link between oxic in situ production of $CH_4$ and algal dynamics (i.e., photosynthesis and respiration rates) (Bogard et al., 2014). Oversaturation of $CH_4$ in oxic surface waters of lakes can also result from $CH_4$ release from littoral sediments in combination with horizontal transport to the open water. The relative importance of both processes is under debate (Bogard et al., 2014; Encinas Fernández et al., 2016; Grossart et al., 2011; Peeters and Hofmann, 2021). The second process also explains better the higher $CH_4$ concentrations observed in shallow zones compared to deep waters of lakes (Peeters and Hofmann, 2021). For the Kolyma River, we propose that the oversaturated $CH_4$ concentrations located close to the river bank and at confluences with tributaries, and the presence of methanogens, is mainly caused by the lateral release of $CH_4$-rich pore water and soil-borne methanogens. This process might be dominant during permafrost melting and resuspension events, rather than in situ net production of $CH_4$ in oxic surface waters by active methanogens."*

L457: $r^2 = 0.11$ and $0.21$ are rather low correlation coefficients, can you add p values? If p is not significant this needs to be indicated so as to not mislead the reader on the presence of correlation.

We have added the statistical significance of the mentioned correlations between water properties and the absolute abundances of methano-/methylotrophs at stations. The modified paragraph reads as follows:

*"The correlations between the total absolute abundances of archaeal microbial communities against the water properties at stations (Fig. 7) show statistically significant ($p<0.05$) positive linear correlations between T and the abundance of methanogens ($r^2=0.35$, $p = 0.005$) and methano-/methylotrophs ($r^2=0.43$, $p=0.001$) (Figs. 7a and 7b). A statistically significant negative linear correlation was obtained against $\kappa$ ($r^2=0.31$, $p=0.007$) for methanogens and for methano-/methylotrophs ($r^2=0.24$, $p=0.02$) (Figs. 7c and 7d). The $pCH_4$ at stations is also positively correlated to the abundance of methano-/methylotrophs ($r^2=0.22$, $p=0.04$) and methanogens ($r^2=0.11$, $p=0.15$) (Figs. 7e and 7f), however the latter is not statistically significant at $p<0.05$".*

L459-470: The paragraph on DOC concentration comes a bit out of the blue after the microbial analysis, maybe add its separate paragraph? Especially because it is not used afterwards

As mentioned in the comment above, we substituted the results of DOC by those of TOC. Given that there is no clear evidence that TOC (or DOC) played an important role as indicators for the $CH_4$ distribution in the river, and that the TOC data is only available for 7 stations, this paragraph has been shortened and only the correlation results between TOC and $pCH_4$ are presented as shown above.

L585: Some methanogens have been detected in oxic environments before, see Bogard et al., 2014 (Oxic water column methanogenesis as a major component of aquatic CH4 fluxes, Nature).

*Thanks again for this comment. We agree that it would strengthen the manuscript by adding in the revised manuscript a paragraph addressing the possibility of methanogenesis in oxic waters. Detailed information has been mentioned in the response above.*

Typographical corrections:

Since you choose to simplify methane as "$CH_4$" please to do homogenously in the manuscript

*Thanks for this comment, we replaced "methane" by "$CH_4$" in all appearances.*

**References cited in this comment**

Bogard, J. M., del Giorgio, P. A., Boutet, L., Garcia Chavez, M. C., Prairie, Y. T., Merante, A., and Derry, A. M.: Oxic water column methanogenesis as major component of aquatic $CH_4$ fluxes, Nature Communications, 5, 5250, 10.1038/ncomms6350, 2014.

Canning, A., Körtzinger, A., Fietzek, P., and Rehder, G.: Technical note: seamless gas measurements across Land-Ocean Aquatic Continuum - corrections and evaluation of sensor data for $CO_2$, $CH_4$ and $O_2$ from field deployments in contrasting environments, Biogeosciences, 18, 1351-1373, 10.5194/bg-18-1351-2021, 2021a.

Encinas Fernández, J., Peeters, F., and Hofmann, H.: On the methane paradox: transport from shallow water zones rather than in situ methanogenesis is the major source of $CH_4$ in the open surace water of lakes, Journal of Geophysical Research: Biogeosciences, 121, 2717-2726, 10.1002/2016JG003586, 2016.

Grossart, H.-P., Frindte, K., Dziallas, C., Eckert, W., and Tang, K. W.: Microbial methane production in oxygenated water column of an oligotrophic lake, Proceedings of the National Academy of Sciences of the United States of America, 108, 19657-19661, 10.1073/pnas.1110716108, 2011.

Peeters, F. and Hofmann, H.: Oxic methanogenesis is only a minor source of lake-wide diffusive $CH_4$ emissions from lakes, Nature Communications, 12, 1206, 10.1038/s41467-021-21215-2, 2021.